# MEASUREMENT REPORT: WINTERTIME AEROSOL CHARACTERIZATION AT AN URBAN TRAFFIC SITE IN HELSINKI FINLAND

Kimmo Teinilä[1], Sanna Saarikoski[1], Henna Lintusaari[2], Teemu Lepistö[2], Petteri Marjanen[2], Minna Aurela[1], Heidi Hellén[1], Toni Tykkä[1], Markus Lampimäki[3], Janne Lampilahti[3], Luis Barreira[1], Timo Mäkelä[1], Leena Kangas[1], Juha Hatakka[1], Sami Harni[1], Joel Kuula[1], Jarkko V. Niemi[4], Harri Portin[4], Jaakko Yli-Ojanperä[5], Ville Niemelä[6], Milja Jäppi[2], Katrianne Lehtipalo[1,3], Joonas Vanhanen[7], Liisa Pirjola[8,3], Hanna E. Manninen[4], Tuukka Petäjä[3], Topi Rönkkö[2] and Hilkka Timonen[1]

[1]Atmospheric Composition Research, Finnish Meteorological Institute, Helsinki, Finland

[2]Aerosol Physics Laboratory, Physics Unit, Faculty of Engineering and Natural Sciences, Tampere University, Tampere, Finland

[3]Institute for Atmospheric and Earth System Research/Physics, Faculty of Science, University of Helsinki, Finland

[4]Helsinki Region Environmental Services Authority (HSY), Helsinki, Finland

[5]Vaisala Oyj, Helsinki, Finland (currently University Association of South Ostrobothnia)

[6]Dekati Ltd, Kangasala, Finland

[7]Airmodus Ltd, Erik Palménin aukio 1, FI-00560 Helsinki, Finland

[8]Department of Automotive and Mechanical Engineering, Metropolia University of Applied Sciences, Vantaa, Finland

*Correspondence to*: Kimmo Teinilä (kimmo.teinila@fmi.fi)

**Abstract.** Physical and chemical properties of particulate matter and concentrations of trace gases were measured at an urban site in Helsinki, Finland for five-weeks to investigate the effect of wintertime conditions on pollutants. The measurement took place in a street canyon (Traffic Supersite) in January–February 2022. In addition, measurements were conducted in an urban background station (UB Supersite, SMEAR III, located approx. 0.9 km from the Traffic Supersite). Measurements were also made using the mobile laboratory. The measurements were made driving the adjacent side streets and the street along the Traffic Supersite. A source apportionment was performed for the Soot Particle Aerosol Mass Spectrometer measurements to identify organic factors connected to different particulate sources. Particle number concentration time series and the Pollution Detection Algorithm were used to compare local pollution level differences between the sites.

During the campaign three different pollution events were observed with increased pollution concentrations. The increased concentration during these episodes were due to both trapping of local pollutants near the boundary layer and long-range and regional transport of pollutants to Helsinki metropolitan area. Local road vehicle emissions increased the particle number concentrations, especially sub-10 nm particles, and long-range and regional transported aged particles increased the PM mass and particle size.

## 1 Introduction

Exposure to increased particulate and gaseous pollutants can have adverse health effects on human health (Atkinson et al., 2014). Especially exposure to elevated concentrations of particulate matter (PM) is estimated to cause 3.3 million premature deaths/year on the global level (Lelieveld et al., 2015). Fine particles ($D_p < 2.5$ μm) are harmful since they can be transported deep into the human respiratory tract (Zanobetti et al., 2014). Especially ultrafine particles ($D_p < 0.1$ μm) may cause serious health problems since they can enter even deeper to the respiratory tract (Schraufnagel, 2020) and their concentration can be very high near local sources e.g. near heavily trafficked streets and highways or in street canyons (Pirjola et al., 2017; Trechera et al., 2023).

In earlier studies it has been shown that main local anthropogenic sources in Helsinki metropolitan area are direct vehicular emissions, road dust, and residential wood burning (Aurela et al., 2015; Carbone et al., 2014; Järvi et al., 2008; Saarikoski et al., 2008; Savadkoohi et al., 2023). Especially the concentration of ultrafine particles can increase near heavily polluted streets and street canyons during the morning and evening rush hours (Hietikko et al., 2018; Lintusaari et al., 2023; Okuljar et al., 2021; Trechera et al., 2023). In addition to local sources, long-range or regional transportation increases pollutant concentrations in Helsinki metropolitan area occasionally (Niemi et al., 2004, 2005, 2009). Local pollutants, mainly vehicle exhaust emissions, increase the particle number concentration due to the increased concentration of ultrafine particles (Rönkkö et al., 2017). In contrast, long-range or regionally transported particles increase the concentration of particulate mass due to the larger size of aged aerosol particles. Lung deposited surface area (LDSA) is used to predict the health effects of particulate matter related to the particle deposition in the lung alveoli. Increased LDSA concentrations are connected to both increased number concentrations of ultrafine particles and increased particle size during the episodes with long-range or regional transported aerosol (Kuula et al., 2020; Lepistö et al., 2023a; Liu et al., 2023).

In addition to temporal and diurnal variation of pollutant sources, local meteorology affects the pollutant concentrations in Helsinki metropolitan area. Especially wind speed may either decrease or increase pollutant concentrations. Concentrations of gaseous and particulate pollutants from nearby sources like motor vehicle exhausts decrease together with increasing wind speed due to more effective ventilation (Teinilä et al., 2019). On the other hand, concentration of coarse particles ($D_p > 2.5$ μm) may increase due to the resuspension of street dust during windy periods. Volatile organic compounds emitted from motor vehicle engines can produce secondary organic aerosol (SOA, e.g., Gentner et al., 2017). Cold periods during wintertime cause stagnant conditions with low mixing height trapping the pollutants in the boundary layer and increasing their concentrations. Snow cover, rain, and wet snow inhibit the resuspension of street dust during wintertime.

A five-week intensive campaign at a Traffic Supersite was conducted during winter 2022 in Helsinki, Finland. The aim of the study was to investigate the role of wintertime conditions in aerosol formation and precursor gases, black carbon (BC) emissions, emission sources, and their influence on particles' physical and chemical properties. During wintertime, temperature inversion episodes cause traffic related pollutants to be trapped on the boundary layer hindering the mixing and dilution of pollutants. Also, photochemical reactions are minimal during wintertime and the contribution of biogenic emissions is limited. Dispersion of street canyon emissions were also studied using mobile measurements with the Aerosol and Trace-gas mobile laboratory (ATMo-Lab) by Tampere University near and at the measurement site. Particle physical and chemical properties were measured also at an urban background station (UB Supersite) during the campaign.

## 2    Experimental

### 2.1    Measurement sites

#### 2.1.1    Traffic Supersite, Mäkelänkatu

The Traffic Supersite station was the principal measurement site during the winter campaign. The Traffic Supersite station is an urban measurement station operated by the Helsinki Region Environmental Services Authority (HSY), located in a street canyon on the street Mäkelänkatu (60.19654 N, 24.95172 E) in Helsinki (Fig. 1). At the Traffic Supersite, a continuous monitoring of urban air quality together with the detailed measurements of particle physical and chemical properties is taking place. An additional measurement container was placed next to the Traffic Supersite station for installing additional measurement devices during the intensive campaign.

Mäkelänkatu street, next to the Traffic Supersite station, consists of six lanes, two rows of trees, two tram lines and two pavements, resulting in a total width of 42 m in the vicinity of the Traffic Supersite station. More detailed descriptions of the site and its air flow patterns are found in Hietikko et al., 2018, Kuuluvainen et al., 2018, and Olin et al., 2020, During the measurement campaign, the average number of vehicles driving along the street was 17 000 per day during workdays, and the share of heavy-duty vehicles was 10 % (statistics from the City of Helsinki).

#### 2.1.2    Urban background supersite, SMEAR III, Kumpula

The SMEAR III measurement station is an urban background supersite (UB Supersite) located in the Kumpula campus area (Fig. 1, Järvi et al., 2009) There is one main road nearby, approx. 150 m from the station, with a daily traffic load of approx. 50 000 vehicles also containing a considerable number of heavy-duty vehicles. However, the UB Supersite is less affected by the local traffic compared to the Traffic Supersite because of the markedly longer distance to the main road. The site is also affected by local residential wood combustion emissions, especially during the winter months. A more detailed description of the UB Supersite surroundings is given in Järvi et al. (2009).

At the UB Supersite, aerosol particle physical and chemical properties and trace gases are continuously measured. During the intensive campaign additional instrumentation was placed at the UB Supersite (see below). The measurements at the UB Supersite were used to get information on the aerosol and trace gas properties in urban background areas.

#### 2.1.3    Rural site, Luukki

Luukki measurement station operated by the HSY is a Helsinki metropolitan area background station situated in a clean background area (20 km from the Traffic Supersite) with no major local pollution sources nearby. The increased concentrations of $PM_{2.5}$ and BC due to long-range or regional transport of particulate matter are typically observed at the Luukki measurement station together with the measurement stations inside the city centre area. The concentrations of $PM_{2.5}$ and BC at the Luukki measurement station were measured using Fidas 200 (Palas GmbH) and Multi-Angle Absorption Photometer (MAAP, Thermo Electron Corporation) instruments.

### 2.2    Instrumentation

#### 2.2.1    Stationary measurements at Traffic Supersite and UB Supersite

*SP-AMS*

The chemical composition of aerosol particles was studied with a Soot Particle Aerosol Mass Spectrometer (SP-AMS,
Aerodyne Research Inc; Onasch et al., 2012) at the Traffic Supersite. Shortly, the AMS consists of a particle sampling
inlet, a particle-size chamber, and a particle composition detection system. After entering through critical orifice and
aerodynamic lenses, particles are size-separated in a Time-of-Flight (ToF) chamber and vaporized either on a tungsten
plate (600 °C) or with an intracavity Nd-YAG-laser (1064 nm) which both were used in this study. The resulting species
are ionized by electron impaction (70 eV) and detected with Time-of-Flight mass spectrometry. The size range covered
by the SP-AMS is achieved with the aerodynamic lens system   which exhibits nearly 100 % transmission efficiency from
approximately 70 to 500 nm (aerodynamic diameter, e.g. Canagaratna et al., 2007; Jayne et al., 2000).  In addition to non-
refractory species like organic aerosol (OA), sulphate, nitrate, ammonium and chloride, the SP-AMS also measures
refractory black carbon (rBC) as well as other refractory particulate material (e.g., metals). However, rBC concentrations
are not shown in this paper since the BC size emitted by traffic is partially below the transmission efficiency of the SP-
AMS.
In this study, the SP-AMS was operated with a 60 s time-resolution of which half was measured in mass spectrum mode
(bulk mass concentrations) and half in Particle Time-of-Flight (PToF) mode (mass size distributions). Composition
dependent collection efficiency (CE) was calculated based on Middlebrook et al., (2012). The effective nitrate response
factor and relative ionization efficiency (RIE) of ammonium ($RIE_{NH4}$: 4), and sulphate ($RIE_{SO4}$: 0.9) were determined by
calibrating the instrument by using dried size-selective ammonium nitrate and ammonium sulphate particles.  The default
RIE values for organic aerosol (1.4) and chloride (1.3) were used. The SP-AMS data was analysed using a standard AMS
data analysis software (SQUIRREL v.  1.63B and PIKA v. 1.23B) within Igor Pro 6 (Wavemetrics, Lake Oswego, OR).
For the elemental analysis of OA an Improved-Ambient method was used (Canagaratna et al., 2015). The sources of OA
were investigated by Positive Matrix Factorization (PMF, Paatero, 1999) using the SoFi Pro software package (version
8.4.0). It employs a multilinear engine (ME-2) as a PMF solver (Canonaco et al., 2013).
The uncertainties of the AMS measurement arise from several factors. One source of uncertainty is the used effective
nitrate response factor which is determined by the calibration of the AMS. Also, the use of default RIE for the calculation
of total OA concentration is a source of uncertainty as a single RIE value for organics may not represent thousands of
different organic compounds found in particles. The fact that the lower size range of the AMS is 50 nm has only a minor
effect on the measured concentrations since the majority of $PM_1$ mass is in particles above this size. The calculation of
CE based on the chemical composition of measured aerosol is an additional source increasing the uncertainty as it uses
nitrate, ammonium and sulphate concentrations in the calculation. The overall uncertainty of the AMS measurements can
be estimated to be about 20–30 %.
*ACSM*
Chemical composition of particulate matter ($PM_1$) was measured continuously using an Aerosol Chemical Speciation
Monitor (ACSM, Aerodyne Research Inc., Ng et al., 2011) at the UB Supersite. The ACSM characterises non-refractory
aerosol species (total organics, sulphate, nitrate, ammonium, and chloride) with a time resolution of approximately 30
min. The ACSM measures particles that pass through the aerodynamic lens that is similar to the aerodynamic lens used
in the AMS. The flow into the ACSM (controlled by critical orifice) was roughly 0.1 l min$^{-1}$, but in addition bypass flow
of 3 l min$^{-1}$ was used to get particles efficiently close to the inlet of the ACSM. A cyclone (URG, URG-2000-30ED) was

used before the ACSM to remove particles larger than 2.5 μm (aerodynamic diameter) to prevent clocking the critical orifice. The uncertainties related to the ACSM measurements are like those described for the AMS above. However, the measured concentration of chloride and ammonia were very low during the campaign so especially for these two components the measurement uncertainty is clearly higher. The estimation for the uncertainties is 50 % for ammonia and >50 % for chloride.

*GC-MS/FID*

Volatile organic compounds (VOCs) and intermediate volatile organic compounds (IVOCs) containing 6 to 15 carbon atoms were measured with 1-hour time resolution using an in situ Thermal Desorption-Gas Chromatograph-Mass Spectrometer (TD-GC-MS) at the Traffic Supersite. Quantified compounds included 10 terpenoids, 15 alkanes, 20 aromatic hydrocarbons, 4 oxygenated aromatic hydrocarbons, and 8 polycyclic aromatic hydrocarbons (PAHs) (Table S1). The system consisted of TurboMatrix 350 connected to an online sampling accessory (TD), Clarus 680 (GC), and Clarus SQ 8 T (MS) all manufactured by PerkinElmer. GC column used was an Elite-5MS 60 m x 0.25 mm (i.d.), film thickness 0.25 μm (PerkinElmer). Sample was collected to the TD's Tenax-TA & Carbopack B dual absorbent cold trap which was kept at 20 °C. Sampling was done approx. 3 m from street level. The inlet was 1/8-inch FEP line, and the outside portion was heated to be around 30 °C. Ozone was removed from the sample flow by guiding the flow through a 1/8-inch stainless steel tube heated to 120 °C. A flow of 300–800 ml min$^{-1}$ was kept through the inlet from which the TD collected 30–45 min samples with a flow of 40 ml min$^{-1}$. More detailed description of the system and method can be found in Helin et al. (2021).

Additional sorbent tube samples were collected at the UB Supersite. Samples were collected to Tenax-TA & Carbopack B dual absorbent tubes via modified Sequential Tube Sampler (STS 25 Unit, PerkinElmer). Main modifications for the sampler were an upgrade to the sampling pump and the exchange of rain cover from stainless steel to PFA. The STS unit consists of a carrousel that rotates on a timer placing the tubes to the slot for active sampling. The carrousel holds 24 tubes at a time and sampling time was set for 4 h making sampling sets approx. 4 days long. Sampling flow was kept around 100 ml min$^{-1}$. Tubes were then analysed in the laboratory with a similar TD-GC-MS setup as described above for in situ samples.

Non-methane hydrocarbons (NMHCs) containing 2–5 carbon atoms were sampled with stainless steel vacuum canisters at the Traffic Supersite. The flow from ambient to the vacuum of the canisters were restricted by a critical orifice making sampling time 24 h. The canister walls were coated with silcosteel. Before analysis canisters were over pressurized with pure nitrogen (99.9999 %). From the pressurised canisters samples were collected to the cold trap of the Markes international Unity 2 via AirServer addon. The system had a Dean switch with dual column and detector setup. First column was DB-5MS 60 m x 0.25 mm (i.d.), film thickness 1 μm (Agilent), and after that the most volatile compounds (C2–C5) were directed to a second column CP-Al2O3/KCl 50 m x 0.32 mm (i.d.), film thickness 5 μm (Agilent) via the Dean switch. C2–C5 compounds were analysed with a Flame Ionization Detector (FID) and rest with the MS. The setups GC/FID was Agilent 7890A and the MS Agilent 5975C. The detection limits for the different VOCs varied between 0.2 and 16 ng m$^{-3}$. Average uncertainty was 2.8, 25 and 18 ng m$^{-3}$ for terpenoids, aromatic compounds and C6-C15 alkanes, respectively.

*CI-API-TOF-MS*

At the Traffic Supersite gaseous sulphuric acid ($H_2SO_4$) was sampled and measured in the same way as described in Olin et al. (2020) with a Nitrate-ion based Chemical-Ionization Atmospheric-Pressure-Interface Time-of-Flight Mass Spectrometer (nitrate CI-API-TOF-MS, Aerodyne Research Inc. USA and Tofwerk AG Switzerland). A high flow rate of outdoor air was pulled in through the roof into the container with a vertical probe and a fan. A vacuum pump pulled in a partial flow of $12\ l\ min^{-1}$ from the vertical probe and $19.01\ l\ min^{-1}$ of reagent flow of which $19\ l\ min^{-1}$ was sheath air and $0.01\ l\ min^{-1}$ flowed over a reservoir of liquid $HNO_3$. The air for the reagent flow was drawn from inside the container and filtered with a HEPA-filter. A diaphragm pump also pushed to provide sufficient flow. The flows were combined and proceeded through an X-ray source which ionized $HNO_3$ vapor into $NO_3^-$ ions. The instrument pulled in approximately $0.1\ l\ min^{-1}$ through a critical orifice and the excess flowed through an active carbon filter, HEPA-filter, and vacuum pump to outside of the container. $NO_3^-$ ionized $H_2SO_4$ into $HSO_4^-$ by receiving a proton. Inside the instrument quadrupoles directed the sample through differential pumping stages onto a detector inside a Time-of-Flight chamber. The CI-API-TOF-MS data is not shown in this paper.

*NAIS*

Two Neutral Cluster and Air Ion Spectrometers (NAIS, Airel Ltd, Manninen et al., 2016; Mirme and Mirme, 2013) were used to measure size and mobility distributions of aerosol particles and air ions at the Traffic Supersite station (NAIS 5-27) and at the UB Supersite station (NAIS12). Air ions of both polarities in the electric mobility range from 3.2 to 0.0013 $cm^2\ V^{-1}\ s^{-1}$ (~0.8–40 nm in mobility diameter) and the distribution of aerosol particles in the size range from ~2 nm to 40 nm were measured with a maximum time resolution of 1 s. Both instruments sampled via horizontal copper inlets with the sample flow rate of $\sim54\ l\ min^{-1}$. The total particle concentrations measured by the NAISs have been observed within ±50 % of the reference CPC concentration at 4–40 nm sizes (Asmi et al., 2009).

*$SO_2$ Analyser*

Enhanced Trace Level $SO_2$ Analyser (Thermo Scientific™, Model 43i-TLE) was employed at the measurement container next to the Traffic Supersite between January 29 and February 22. Data was collected in 20 s time resolution with the instrument frow rate of $0.5\ l\ min^{-1}$. $SO_2$ data is not shown in this paper.

*CPCs*

Two condensation particle counters (CPC), TSI model 3756 (UB Supersite) and Airmodus model A20 (Traffic Supersite) were used to measure particle number concentration time series. The TSI 3756 has a particle concentration range up to $300\ 000\ cm^{-3}$ and size range down to 7 nm ($Dp_{50}$), and the maximum detectable particle size > 3 μm. Inlet flow rate of $1.5\ l\ min^{-1}$ was used. The Airmodus A20 was used with a bifurcated flow diluter (dilution ratio 8.5), which expands concentration range up to $250\ 000\ cm^{-3}$ in single particle counting mode. The particle size range measured was from 5.4 nm ($Dp_{50}$) to 2.5 μm. The uncertainties of the CPCs are typically within 10% concentrations of the ambient aerosol ranging from a few thousands up to $100\ 000$ particles $cm^{-3}$ (Schmitt et al., 2020). In both CPC types, butanol (n-Butyl alcohol) was used as a working fluid and data was collected at 1 min time resolution.

*nCNC with AND*
An Airmodus Nanoparticle Diluter (AND) (Airmodus Ltd, Lampimäki et al., 2023) was used to dilute sample air upstream
of a nano-Condensation Nucleus Counter (nCNC) system (Airmodus Ltd, Vanhanen et al., 2017) at the Traffic Supersite.
The nCNC measures the particle activation size distribution between ca. 1 and 4 nm by scanning the cut-off size. The
default dilution factor of 5 was used by using dry compressed air in the dilution, which also allowed drying of the sample
to < 30 % relative humidity (RH). In the preset study the Ion Precipitator (IOP) voltage (1 kV) of AND was sequentially
switched on and off with a custom-made MATLAB based program. IOP can be used to scavenge ions at the mobility
diameters below ~8 nm, while the larger (> 10 nm) particles are passing through the IOP with the 50 % cut-off size around
9 nm. Thus, the IOP mode could provide additional information on the charged fraction of recently formed particles or
clusters. The nCNC/AND data has not been shown in this paper.

MAAP
Black carbon concentration was measured using a Multi-Angle Absorption Photometer (Thermo Electron Corporation,
Model 5012, Petzold and Schönlinner, 2004) at the Traffic Supersite and at the UB supersite stations. The MAAP
determines the absorption coefficient ($\sigma$AP) of the particles deposited on a filter by a simultaneous measurement of
transmitted and backscattered light. The $\sigma$AP is converted to BC mass concentrations by the instrument firmware using a
mass absorption cross section of 6.6 m$^2$ g$^{-1}$ (Petzold and Schönlinner, 2004). The flow rate of the MAAP at the Traffic
Supersite was 11 l min$^{-1}$ and 5 l min$^{-1}$ at the UB Supersite. Both MAAP instruments measured with one minute time
resolution and cyclone/PM$_1$ inlet was used to cut-off particles above 1 μm. At traffic Supersite the measured BC
concentration was most of the time above the detection limit of the MAAP so the measurement uncertainty is mostly due
to the uncertainties in the sampling like particle losses in the sampling lines. The uncertainty of the MAAP results can be
estimated be around 10–15 %. At UB Supersite the measured BC concentration was more frequently near or below the
detection limit. This can cause larger uncertainties for the BC measurements at UB Supersite.

*AE33*
An AE33 dual spot aethalometer (Magee Scientific, Slovenia) was used to measure the aerosol light absorption and
corresponding carbon mass concentrations at the Traffic Supersite and at the UB Supersite. The AE33 measures at seven
different wavelengths between 370 and 950 nm (Drinovec et al., 2015; Hansen et al., 1984). The flow rate of the AE33
was 5 l min$^{-1}$ and the used filter tape was PTFE-coated glass fibre filter (no. M8060). The cut-off size of the sample was
1 μm at both station, and it was achieved using a sharp cut cyclone (Model SCC1.197, BGI Inc., Butler, NJ, USA). The
data from the AE33 instrument is not used in this paper.

*AQ Urban*
The alveolar LDSA concentration of aerosol particles between 10 and 400 nm was measured with the Pegasor AQ$^{TM}$
Urban instrument (Pegasor Ltd., Finland) at the Traffic Supersite and at the UB Supersite (Kuula et al., 2020). The
measured LDSA concentration was typically above the detection limit of the AQ Urban instrument (1 μm$^2$ cm$^{-3}$).

*DMPS*
A Differential Mobility Particle Sizer (DMPS) was used to measure particle size distributions from 11 to 800 nm (Traffic
Supersite) and from 3 to 800 nm (UB Supersite) using a Vienna type Differential Mobility Analysers and an Airmodus
A20 model CPC (Traffic Supersite) and TSI 3025 CPC (UB Supersite). The particle number size distributions from 20 to
200 nm determined by the mobility particle size spectrometers are typically within an uncertainty range of around $\pm 10$
%, while below and above this size range the uncertainty increases. For particle sizes above 200 nm, 30 % uncertainty
has been reported (Wiedensohler et al., 2012).

*Picarro*

Gas analyser for carbon monoxide (CO), carbon dioxide ($CO_2$) and methane ($CH_4$) at both sites was a Picarro G2401
manufactured by Picarro Inc. (Santa Clara, CA). It also measures water vapor concentration, based on which it calculates
dry concentrations for the other components.  The instrument is based on cavity ringdown spectroscopy (CRDS), in which
long optical path length allows measurements with high precision and stability using near-infra-red laser sources.
The Kumpula instrument close to the UB Supersite took its sample air from the roof of the five-store Finnish
Meteorological Institute's building ca. 30 m above the ground. The sample air was dried with a Nafion dryer run in reflux
mode. The Traffic Supersite Picarro was run with non-dried sample air. Both instruments were calibrated with WMO/CCL
(World Meteorological Organization/Central Calibration Laboratory) traceable gases. The measured $CO_2$ and $CH_4$
concentrations were above the detection limit of Picarro at both sites. The uncertainty of these two gases is low (10 %)
but for the measured CO concentrations the uncertainty can be larger.

*Filter sampling and chemical analysis*

In addition to online measurements, PAH filter samples ($PM_{10}$) were collected on daily basis at the Traffic Supersite. Also
12–24 -hour filter samples ($PM_1$) were collected for sugar anhydride (levoglucosan, mannosan and galactosan) and
elemental carbon/organic carbon (EC/OC) analyses at the Traffic Supersite. Quartz fibre filters (PALL, Tissuquartz 2500-
QAT-UP, NY, USA) were used as sampling substrates for the $PM_1$ samplings with the flow rate of 20 l $min^{-1}$.

*Other instrumentation*

Concentration of particulate mass ($PM_{2.5}$ and $PM_{10}$) were measured with Fidas 200 (Palas) instrument at the Traffic
Supersite. Concentrations of gaseous compounds were also continuously measured at the Traffic Supersite. APNA 370
(Horiba) instrument was used for measuring the concentrations of $NO_x$ (APNA 370) and $O_3$ (APOA 370), APMA 360
(Horiba) was used to measure the concentration of CO and LI-7000 (LICOR) was used to measure the concentration of
$CO_2$. Particle scattering coefficient was measured with a nephelometer (TSI, model 3610) at the Traffic Supersite. The
measurement devices at the Traffic Supersite are shown in Table S2 and those at the UB Supersite in Table S3.
Back trajectories of air masses arriving to the Traffic Supersite were calculated using the NOOA HYSPLIT model (Rolph
et al., 2017; Stein et al., 2015). The 96-hour back trajectories were calculated for every hour for 200 m above sea level.
Mixing height was calculated using model developed at Finnish Meteorological Institute (MPP-FMI, Karppinen et al.,
2000, detailed description of the mixing height calculation is in the supplement). The data analysis was made using the R
software (R Core Team 2022) and R package openair (Carslaw and Ropkins, 2012). Hourly mean concentrations of the

measured components were used in the following discussion unless otherwise mentioned. The used timestamp for hourly mean concentrations is the end hour and the used datetime is local time.

*PAH analyses*

The concentrations of 6 PAHs (benzo(a)anthracene, benzo(b)fluoranthene, benzo(k)fluoranthene, benzo(a)pyrene, indeno(1,2,3-cd) pyrene, dibenz(a,h)anthracene) were analysed from daily $PM_{10}$ samples using a Gas Chromatograph-Mass Spectrometer (GC-MSMS, Agilepicant 7890A and 7010 GC/MS Triple Quadrupole). For the analysis, the samples were ultrasonic extracted with toluene, dried with sodium sulphate, and concentrated to 1 ml. For chromatographic separation, the HP-5MS UI column (30 m x 0.25 mm i.d., film thickness 0.25 μm) and 2 m pre-column (same phase as analytical column) were used. Helium (99.9996%) was used as a carrier gas with a flow of 1 ml min$^{-1}$. The temperature program started at 60 °C with a 1 min hold, followed by an increase of 40 °C min$^{-1}$ 1 to 170 °C, and 10 °C min$^{-1}$ to 310 °C with a hold of 3 min. Deuterated PAH compounds (Naphthalene-d8, Acenaphthene-d10, Phenanthrene-d10, Chrysene-d12, Perylene-d12, PAH-Mix 31D, Dr. Ehrenstorfer) were used as internal standards and were added to an extraction solvent before extraction. External standards (PAH Mix-137, Polynuclear aromatic hydrocarbons Mix, Dr. Ehrenstorfer) with five different concentration levels were used. In the analysis of benzo(a)pyrene, EN 15549 (2008) standard was followed. Measurement uncertainty was calculated from the validation data (Guide Nordtestin TR537) for the target value (0.1 ng m$^{-3}$) that value was found to be 25 %. The analysis method is accredited (SFS-EN ISO/IEC 17025:2017). The method has been previously described in detail by Vestenius et al., 2011.

*EC/OC analyses*

The concentrations of particulate OC and EC were analysed using a thermal-optical OCEC aerosol analyser (model 5L, Sunset Laboratory Inc., Tigard, OR, US, (Birch and Cary, 1996)). The thermal analytical technique splits carbon into fractions according to their volatility. In the first stage, OC is desorbed from the quartz fibre filter through progressive heating under a pure He stream. However, a fraction of OC may char and form pyrolyzed OC during that stage. In the second phase, the sample is heated in temperature steps under a mixture of 98 % He–2 % $O_2$ (HeOx phase), during which pyrolyzed OC and EC are desorbed. To correct the pyrolysis effect, the analyser measures the transmittance of a 658 nm laser beam through the filter media. The split point, which separates OC and pyrolyzed OC from EC, is determined as a point when the laser signal returns to its initial value. After being vaporised in several temperature steps, OC, EC and pyrolyzed OC are catalytically converted first to $CO_2$ and then to $CH_4$, which is quantified with a flame ionisation detector. At the end of each analysis, a fixed volume of calibration gas (5 % $CH_4$ in helium) is injected into the instrument to correct possible variations in the analyser's performance. In this study EUSAAR-2 protocol was used (Cavalli et al., 2010). The uncertainties of the EC/OC analysis are ~15 % (Cavalli et al., 2023).

*Sugar anhydride analyses*

The concentration of monosaccharide anhydrides (levoglucosan, mannosan and galactosan) was analysed from the $PM_1$ samples using a High-Performance Anion-Exchange Chromatography-Mass Spectrometry (HPAEC-MS). The HPAEC-MS system consists of a Dionex ICS-3000 ion chromatograph coupled with a quadrupole mass spectrometer (Dionex MSQ). The HPAEC-MS system had 2 mm CarboPac PA10 guard and analytical columns (Dionex) and potassium hydroxide (KOH) eluent. The used ionization technique was electrospray ionization. The analytical method is similar than described in Saarnio et al. (2010), except that the used internal standard was methyl-β-D-arabinopyranoside. A 1

cm² punch of the quartz fibre filter was extracted into 5 ml of MQ water with internal standard concentration of 100 ng
ml⁻¹ and the HPAEC-MS was utilized for determination of MA: s at m/z 161. The uncertainty of the analyses was typically
10–15 % and even larger (25 %) when the analysed concentration was low.

2.2.2     ATMo-Lab measurements
In addition to the stationary measurement stations, the Aerosol and Trace-gas mobile laboratory by Tampere University
was utilized in both stationary and mobile measurements between 18 January and 16 February 2022. The focus of the
ATMo-Lab measurements was to understand how the effects of road traffic vary in the studied street canyon compared
to more open sections of the same road. Also, the aim was to study the dispersion of road traffic emissions to the adjacent
streets in a built environment. Stationary measurements were conducted on the kerbside, next to the Traffic Supersite and
along a side street (Anjalantie, 60.197725 N, 24.957364 E) nearby. Mobile measurements included a park and a street
canyon section of the main street (Mäkelänkatu) along which the measurement stations were located and its side streets
with apartment buildings. Stationary measurement locations along with the driving route are presented in Fig. 1.
Measurements were conducted daytime between 6:30 and 19:30, because the effects of traffic were clearest during that
time. Measurement setup inside the Aerosol and Trace-gas mobile laboratory is shown in Fig. S1.


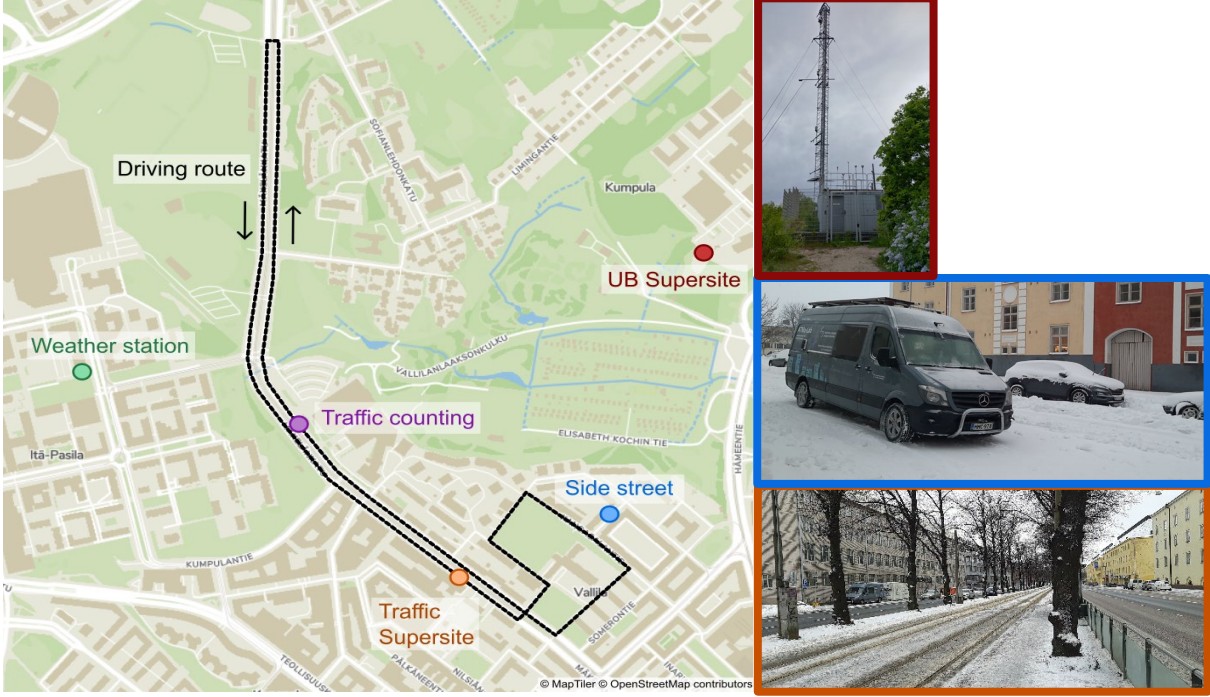


**Figure 1.** Stationary measurement locations and the driving route of the Aerosol and Trace-gas mobile laboratory. The Traffic Supersite
is showed in the map as an orange balloon with a figure right bottom. The side street is showed as a blue balloon and the figure is right
middle. The UB Supersite is showed as a red balloon and with a figure right top.

The sample air was taken from an inlet located above the ATMo-Lab's windscreen and distributed to the instruments at
the back of the van. The risk of self-sampling during driving measurements was minimal as the exhaust pipe is at the rear
end of the van. The van itself is Euro VI compliant. Measurement setup inside the ATMo-Lab is shown in Fig. S1. Key
measurement target of the ATMo-Lab measurements was ultrafine particles. Particle number concentrations were

measured using a Condensation Particle Counter Battery (CPCB) and an Electrical Low-Pressure Impactor (ELPI+, Dekati Ltd). The CPCB consisted of a combination of a Particle Size Magnifier (PSM) and a Condensation Particle Counter (CPC) in parallel with four CPCs with different cut-off sizes. Working principle of the PSM is described in Vanhanen et al. (2011). Exact CPCB instruments were A11 nCNC (combination of PSM and CPC, Airmodus Ltd), CPC 3756 (TSI Inc), CPC 3775 (TSI Inc), CPC A20 (Airmodus Ltd), and CPC A23 (Airmodus Ltd). Respectively, total particle number concentrations were simultaneously measured for size ranges > 1.3 nm, > 2.5 nm, > 4 nm, > 10 nm, and > 23 nm with a time resolution of one second. Sample air was diluted before entering to the CPCB using a bifurcated flow diluter, including a static mixer, with dilution ratio of 17.

The ELPI+ measured the particle number size distribution with its 14 impactor stages in a size range from 6 nm to 10 μm. The operation principle of the ELPI is described in (Keskinen et al., 1992) and (Marjamäki et al., 2000), and the calibration of the renewed ELPI+ is presented in Järvinen et al., 2014. The ELPI+ was also used to measure particle lung deposited surface area (LDSA) and mass (PM). The stage-specific conversion from electric current data of the ELPI+ to LDSA concentration enable measurement of the LDSA concentration and size distribution with the whole ELPI+ size range (Lepistö et al., 2020). LDSA was also measured by a sensor type device Partector (Naneos particle solutions GmbH). Furthermore, $PM_{2.5}$ concentrations were calculated by integrating over the particle number size distribution measured with the ELPI+ assuming spherical particles with unit density. Similar assumptions were made with particle number and LDSA size distributions.

Non-volatile particle number was measured with two prototype instruments originally developed for renewed demands of vehicle inspection: a Mobile Particle Emission Counter (MPEC+, Dekati Ltd) and a Pegasor sensor (Pegasor Ltd). Latter sampled first from the roof of the ATMo-Lab (18 January to 2 February) after which it was also connected to the main line (2 February to 16 February). A combination of a thermodenuder followed by a CPC was used as a reference for the prototype instruments. The thermodenuder model was the same as in Heikkilä et al. (2009) and Amanatidis et al. (2018). The CPC used was a model CPC 3775 (by TSI Inc) with altered cut-off diameter (4 nm, 10 nm, or 23 nm) as the cut-off size was changed twice during the measurements. The cut-off size was changed by altering the condenser temperature of the CPC according to a laboratory calibration.

For the analysis of particles' chemical composition and especially black carbon, the ATMo-Lab setup included a SP-AMS and an AE33 aethalometer like the instruments described in the stationary measurements section. For the driving measurements the SP-AMS menu was switched from the 60 s time-resolution (30 s mass spectrum mode + 30 s PToF-mode) to a 24 s time-resolution operation mode in which 14 s was measured in a mass spectrum mode and 10 s in a PToF-mode. From the gaseous compounds, the ATMo-Lab was equipped to measure $CO_2$ (LI-7000, LI-COR Corp) and NO (Model T201, Teledyne Technologies Inc.).

In stationary measurement locations, particles were collected with and without thermal treatment on holey-carbon grids by a flow-through sampler. This was done for elemental composition and morphology study with a (Scanning) Transmission Electron Microscope (S/TEM) accompanied by an energy-dispersive spectrometry (EDS).

# 3 Results

## 3.1 General description of the measurement campaign.

### 3.1.1 Meteorology

The intensive campaign took place between 17 January and 22 February 2022, which is typically the coldest winter period with a minimum amount of sunlight in Helsinki, Finland. Temperature, relative humidity, wind speed and wind direction, measured about 1 km from the Traffic Supersite during the winter campaign, are shown in Fig. 2. Mean temperature during the measurement campaign was -1.4 $^\circ$C (range $-10.0$–2.9 $^\circ$C) and mean relative humidity was 89 % (range 58–100 %). The temperature was most of the time near 0 $^\circ$C. The prevailing wind direction during the measurements was from south to south-east and the mean wind speed was 4.9 m s$^{-1}$ (range 0.59–11.4 m s$^{-1}$). The average mixing height was 408 m (range 58–844). Surface inversion episodes take place during the coldest winter days with low temperature and wind speed causing gaseous and particulate pollutants to be accumulated in the boundary layer (Barreira et al., 2021; Teinilä et al., 2019). Two cold periods together with low wind speed and low mixing height took place during the campaign (Fig. 2) enabling surface inversion episodes. The conditions during the winter campaign (temperature, inversion episodes and variable snow cover) represented typical winter conditions in Helsinki.

In general, local traffic is an important source of gaseous and particulate pollutants at the Traffic Supersite. The air quality in Helsinki is also affected by long-range transported pollution episodes (Niemi et al., 2004, 2005, 2009; Leino et al., 2014; Pirjola et al., 2017) Local wood burning due to heating of detached houses in winter increase air pollutant concentrations in Helsinki. Majority of this wood burning is taking place outside city centre in the residential suburban area (Kangas et al., 2024). During wintertime the long-range and regional transported air masses consist more particulate pollutants connected to biomass burning (Pirjola et al., 2017; Teinilä et al., 2022).

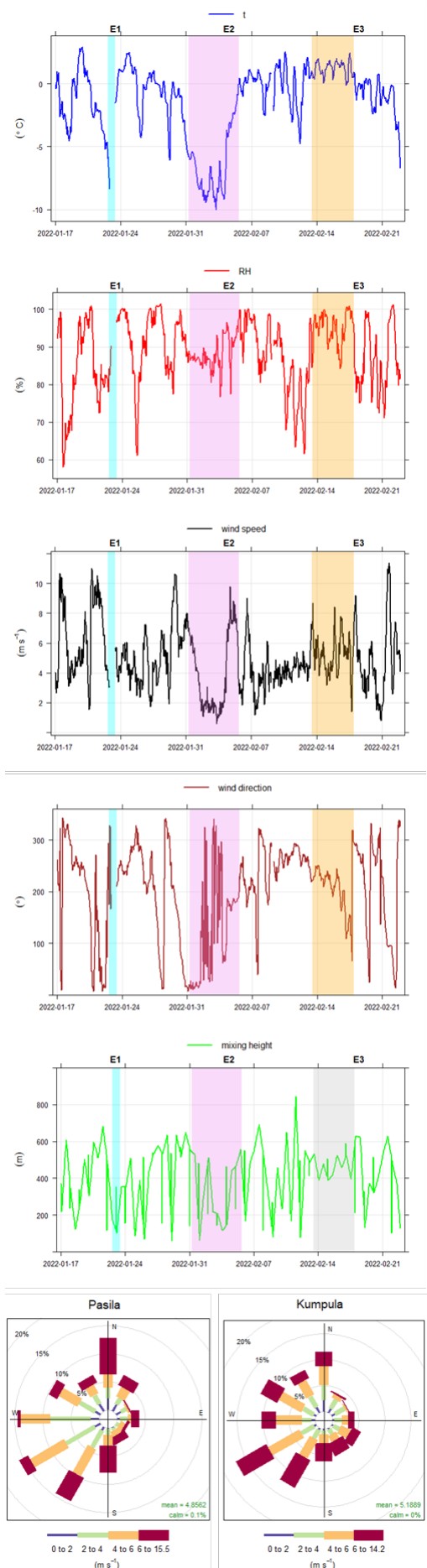

441

**Figure 2.** Temperature, relative humidity, winds speed, wind direction and mixing height during the measurement period measured at Pasila. The three episodes are coloured in the figure. The wind roses at Pasila and Kumpula during the campaign are also shown (bottom figure). The Pasila weather station is about 1 km distance from the Traffic Supersite and the Kumpula weather station is next to the UB Supersite.

### 3.1.2 Traffic frequencies

Traffic frequencies near the Traffic Supersite station are shown in Fig. 3. The exact location for traffic counting is presented in Fig. 1. Traffic frequencies are continuously counted by the City of Helsinki. Inductive loop sensors are installed below the asphalt surface for each driving lane. As the magnetic field of a vehicle passes over the inductive loop, it generates signals that are then recorded. The traffic frequencies were measured about 500 m from the station, but after the measurement point traffic directed to the city centre is divided into two other main streets before the Traffic Supersite. The average number of vehicles passing the Traffic Supersite during workdays was 17 000 per day which is about 40 % less than at the point where traffic frequencies were measured. The traffic frequency at the Traffic Supersite was obtained by manual traffic counting.

Traffic frequency towards the city centre (south) starts to increase around 6:00 and it reaches its maximum between 9:00 and 10:00. The afternoon traffic frequency peak between 17:00 and 18:00 is slightly lower compared to the morning. Traffic frequencies away from the city centre (north) behave opposite trend showing maximum frequencies during afternoon hours. The maximum frequencies out of the city centre are achieved at the same time than those towards to city centre. There are no rush hours during the weekends (Fig. 3) and traffic frequencies starts to slowly increase before noon and show their maximum between 18 and 19 towards both directions. The measurement site is located near to the lines leading towards the city centre (south).

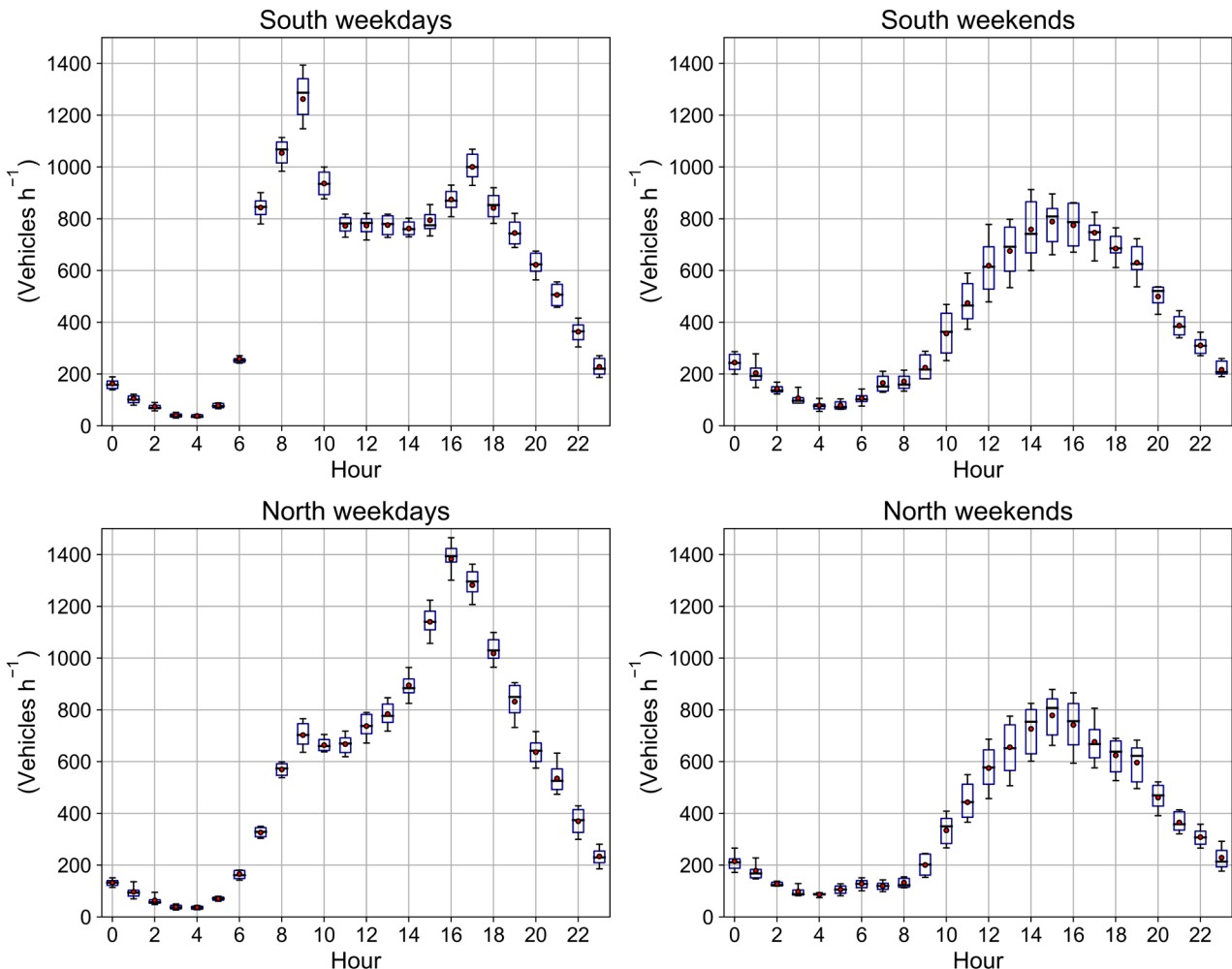

**Figure 3**. Hourly traffic frequencies to south (towards city centre) and to north during workdays and weekends near the Traffic Supersite. The Traffic Supersite station is placed on a pavement on the south traffic side in Mäkelänkatu.

3.1.3     Particle chemical and physical properties during the campaign

The mean concentrations of $PM_{2.5}$ and $PM_{2.5-10}$ were 5.2 and 3.3 µg m$^{-3}$, respectively, during the whole campaign. The maximum 24-hour average concentration of $PM_{2.5}$ was 20.1 µg m$^{-3}$. This 24-hour average exceeds the WHO AQG levels (Air Quality Guideline) during two days during the measurement campaign (WHO AQG level 15 µg m$^{-3}$ for $PM_{2.5}$). The maximum 24-hour average for $PM_{10}$ was 24.7 µg m$^{-3}$ which is clearly below the WHO AQG level of 45 µg m$^{-3}$. The five year average $PM_{2.5}$ concentration between 2015 and 2019 at the Traffic Supersite was 7.2 µg m$^{-3}$ (Barreira et al., 2021). The $PM_{2.5-10}$ concentration was relatively low during the campaign. This is due to rainfall, snowfall, and snow covering the streets during the campaign which inhibited the formation and re-suspension of street dust. Most of the street dust is in coarse particle size, but it is in some degree also in fine particle size range. The lack of street dust episodes in winter explains, at least partly, why the mean $PM_{2.5}$ is also lower than that measured at the Traffic Supersite throughout in years 2015–2019 (Rönkkö et al., 2023b). Some pollution episodes can be observed, and they will be analysed in section 3.1.5.

The mean concentrations of BC, NO, and $NO_2$ were 0.59 µg m$^{-3}$, 12.8 µg m$^{-3}$, and 20.8 µg m$^{-3}$, respectively, at the Traffic Supersite. The 24-hour maximum $NO_2$ concentration during the measurement campaign was 44.7 µg m$^{-3}$ and in 11 days

the WHO AQG level (25 µg m$^{-3}$) was exceed. The mean particle number (PN, DP$_{p50}$: 5.4 nm) concentration at the Traffic
Supersite during the measurement period was 18 093 p cm$^{-3}$ but mean hourly concentrations of PN higher than 80 000 p
cm$^{-3}$ was also measured (Fig. S3). The PN concentration in the ATMo-Lab measurements (Dp$_{50}$: 2.5 nm) next to the
Traffic Supersite was considerably higher than the one measured at the Supersite (Dp$_{50}$: 5.4 nm) (Fig. S4), showing the
effect of road traffic in the emissions of the smallest nanoparticles (e.g. Hietikko et al., 2018; Lintusaari et al., 2023;
Rönkkö et al., 2017). In general, it should be noted that the measured PN concentrations may differ notable depending on
the used instrument cut-off size as also seen when comparing the ATMo-Lab measurements with cut-off sizes 2.5 nm and
10 nm (Fig. S5, Rönkkö et al., 2023a). The mean hourly LDSA concentration at the Traffic Supersite was 13.6 µm$^2$ cm$^{-3}$
during the measurement period (Fig. S2), and the highest hourly mean LDSA concentration during the measurement
period was 62 µm$^2$ cm$^{-3}$.

Mean concentrations of organics, sulphate, nitrate, ammonium, and chloride measured with the SP-AMS (PM$_1$) were 2.0
µg m$^{-3}$, 0.6 µg m$^{-3}$, 0.6 µg m$^{-3}$, 0.4 µg m$^{-3}$, and 0.07 µg m$^{-3}$, respectively. The sum of the concentrations of the measured
chemical components (organic and inorganic species from the SP-AMS and BC from the MAAP) showed a good
correlation coefficient (square of Pearson correlation, R$^2$) against the PM$_{2.5}$ concentrations (0.87).

3.1.4    Volatile organic compounds
The mean concentrations of the continuously measured C6–C15 aromatic hydrocarbons, alkanes, PAHs and terpenoids
were 2.2, 0.94, 0.037 and 0.16 µg m$^{-3}$, respectively, at the Traffic Supersite. Offline samples of C2–C5 NMHCs collected
during the shorter periods showed that light alkanes were the most significant compound group detected (Fig. 4).
However, larger, and more reactive compounds with higher SOA formation potentials are expected to have stronger
impacts on the local chemistry even with lower concentration. Compounds with 6 to 9 carbon atoms were mostly aromatic
hydrocarbons and for higher carbon masses (C10–C11) alkanes and terpenes had a major contribution (Fig. 6). The
contribution of PAHs was very low.  The major contribution of aromatic hydrocarbons was expected due to traffic as a
major local source of VOCs. Higher alkanes (C10–C15), which had highest contribution for IVOCs (Fig 6. C10–C15),
are commonly found especially in diesel emissions (Marques et al., 2022; Wu et al., 2020). Also, terpenoids, which are
traditionally considered as biogenic compounds, had relatively high concentrations during this winter period. Emissions
from the vegetation are expected to be negligible due to cold weather (e.g. Hellén et al. 2021 and Hakola et al. 2023), and
this indicates anthropogenic sources for these compounds. They are commonly found for example in personal care and
cleaning products (Coggon et al., 2021; Steinemann, 2015). In earlier studies terpenoids have been detected also during
wintertime in the urban background air in Helsinki (Hellén et al., 2012).


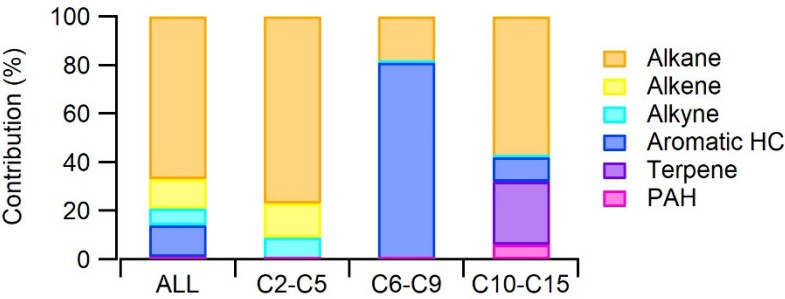

**Figure 4.** Contribution of different compounds groups on measured concentrations during the periods when all VOCs with 2 to 15 carbons (ALL) were measured. C2–C5 = VOCs with 2 to 5 carbons, C6–C9=VOCs with 6 to 9 carbon atoms, and C10–C15=VOCs/IVOCs with 10 to 15 carbon atoms at the Traffic Supersite.

3.1.5    Pollution episodes

During the campaign three episodes with enhanced particulate and gaseous pollutant concentrations were observed. The duration of these episodes is shown in Table 1. The names from E1 to E3 will be used for these episodes in the proceeding chapters.

The timeseries in Fig. 5 show the concentrations of fine ($PM_{2.5}$) and coarse ($PM_{2.5-10}$) particles at the Traffic Supersite, UB Supersite, and Rural site. $PM_{2.5}$ showed elevated concentrations during the three episodes at the Traffic Supersite. $PM_{2.5-10}$ concentration did not show long lasting increase in its concentration during these episodes, but shorter high peaks were observed. The increased concentrations of $PM_{2.5}$ is due to both, of trapping of local pollutants on boundary layer during cold periods and the effect of long-range or regional transport of pollutants at the Traffic Supersite. The increased $PM_{2.5}$ concentration at all sites, also including the Rural site, indicates that long-range or regional transport had an important effect on the air quality in Helsinki Metropolitan area during these episodes. The concentration of $PM_{2.5-10}$, on the other hand, was affected by wind speed and local snow cover or by wet street surface when the temperature was near or above 0 °C (during E3). Also, coarse particles are not typically transported from very long distances. The source of the short-lasting peaks in $PM_{2.5-10}$ concentration may be due to some local activity near the stations (Traffic Supersite and UB Supersite). together with the favourable meteorological conditions as well as non-exhaust emissions from traffic.

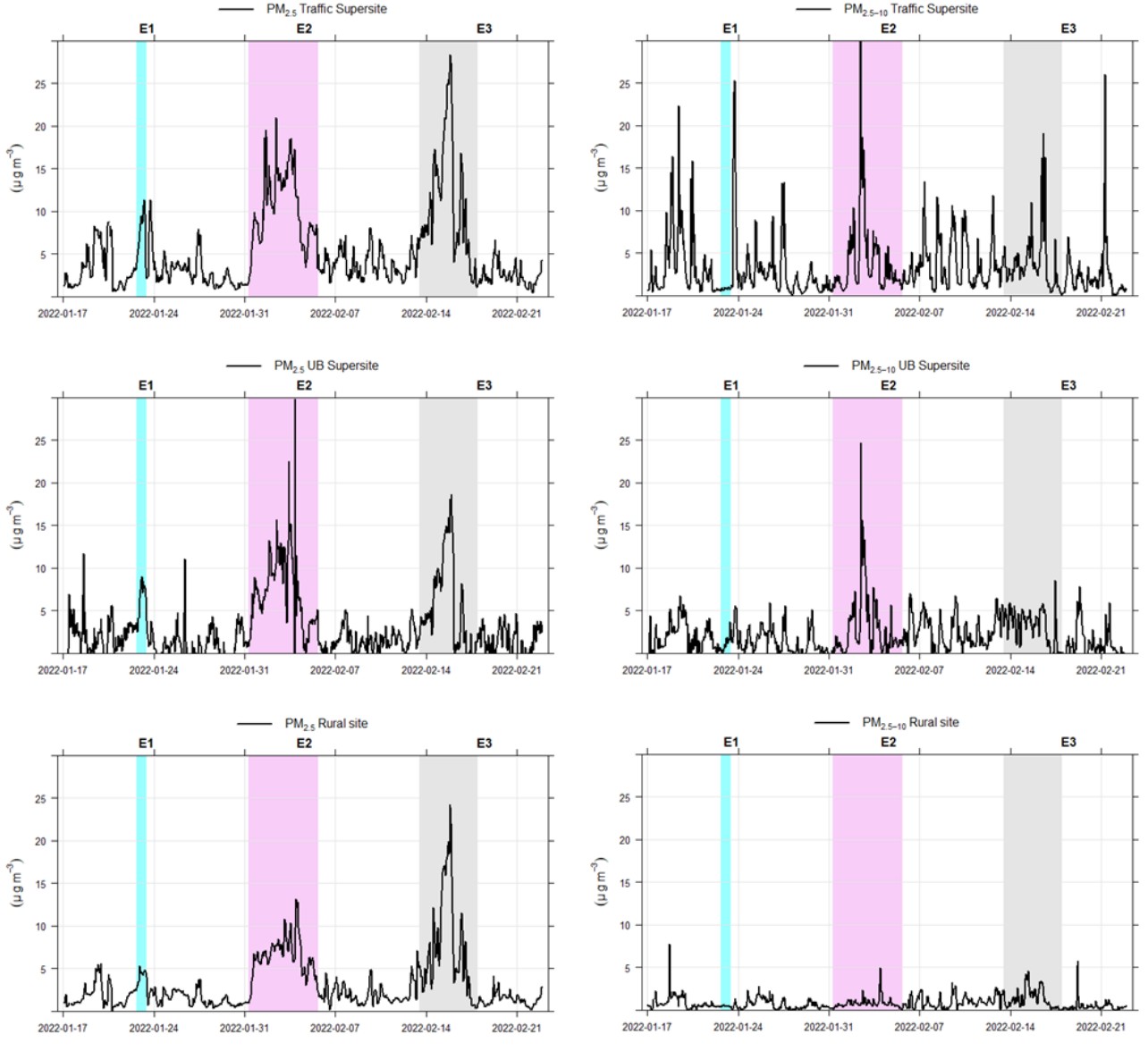

**Figure 5.** Concentrations of PM$_{2.5}$ and PM$_{2.5-10}$ at the Traffic Supersite, UB Supersite and Rural site. The three episodes are colored in the figure.

**Table 1.** Traffic frequencies and average concentrations of measured components during the traffic (averages without episodes) dominated period on workdays and weekends and their averages during three different episodes at the Traffic Supersite.

| Compound | Traffic workdays | Traffic weekends | E1 22.1.2022 15:00 23.1.2022 10:00 | E2 31.1.2022 07:00 5.2.2022 16:00 | E3 13.2.2022 12:00 17.2.2022 23:00 |
|---|---|---|---|---|---|
| ***Traffic frequencies*** *(vehicles per hour)* | 1109 | 767 | 738 | 1126 | 1201 |
| **PN$_{>5nm}$** *(p cm$^{-3}$)* | 19873 | 11970 | 17648 | 23779 | 17117 |
| **LDSA** *(μm$^2$ cm$^{-3}$)* | 12.0 | 8.0 | 16.0 | 23.0 | 16.9 |

| | | | | | |
|---|---|---|---|---|---|
| $PM_{2.5}$ *(µg m$^{-3}$)* | 3.2 | 3.1 | 7.1 | 10.3 | 11.0 |
| $PM_{2.5-10}$ *(µg m$^{-3}$)* | 3.6 | 2.5 | 0.8 | 4.0 | 3.8 |
| $NO$ *(µg m$^{-3}$)* | 13.5 | 4.8 | 4.4 | 25.8 | 11.5 |
| $NO_2$ *(µg m$^{-3}$)* | 22.4 | 12.6 | 21.3 | 27.2 | 22.4 |
| $BC$ *(µg m$^{-3}$)* | 0.46 | 0.29 | 0.97 | 1.17 | 0.90 |
| $CO$ *(ppb)* | 164 | 164 | 202 | 253 | 210 |
| $CO_2$ *(ppm)* | 435 | 432 | 438 | 450 | 439 |
| $CH_4$ *(ppb)* | 2015 | 2021 | 2048 | 2076 | 2056 |
| $O_3$ *(µg m$^{-3}$)* | 46 | 55 | 33 | 25 | 42 |
| *Toluene (µg m$^{-3}$)* | 0.62 | 0.60 | 0.85 | 0.78 | 0.60 |
| *α-pinene (µg m$^{-3}$)* | 0.033 | 0.029 | 0.086 | 0.068 | 0.040 |
| *Particulate organics (µg m$^{-3}$)* | 1.12 | 0.94 | 3.07 | 4.54 | 4.25 |
| *Sulphate (µg m$^{-3}$)* | 0.20 | 0.24 | 1.56 | 2.12 | 0.84 |
| *Nitrate (µg m$^{-3}$)* | 0.26 | 0.19 | 1.78 | 1.58 | 1.48 |
| *Ammonium (µg m$^{-3}$)* | 0.16 | 0.13 | 1.04 | 1.07 | 0.78 |
| *Chloride (µg m$^{-3}$)* | 0.05 | 0.04 | 0.08 | 0.06 | 0.18 |
| | | | | | |
| *HOA (µg m$^{-3}$)* | 0.17 | 0.09 | 0.14 | 0.49 | 0.18 |
| *BBOA (µg m$^{-3}$)* | 0.08 | 0.08 | 0.10 | 0.14 | 0.16 |
| *SV-OOA (µg m$^{-3}$)* | 0.15 | 0.11 | 0.28 | 0.49 | 0.17 |
| *LV-OOA (µg m$^{-3}$)* | 0.27 | 0.25 | 0.14 | 0.33 | 0.66 |
| *LV-OOA-BB (µg m$^{-3}$)* | 0.11 | 0.16 | 1.12 | 1.64 | 1.71 |
| *Tr-OOA (µg m$^{-3}$)* | 0.23 | 0.15 | 0.34 | 0.36 | 0.17 |


The highest increase in their concentrations during the episodes at Traffic Supersite was found for secondary inorganics
(sulphate, nitrate, and ammonium), total organics and BC measured in $PM_1$ (Table 1). α-pinene, known as a SOA
precursor, had also clearly higher concentrations during the periods E1–E3. Concentration of chloride showed a clear
increase only during period E3. Similar increase in the concentrations of BC (Fig. 6), inorganics and organics (Fig. S6)
was seen also at the UB Supersite. The increased $PM_{2.5}$ concentration is thus connected to the formation of secondary
inorganics together with the increased concentration of organics.

Particle number concentrations showed slight increase during the E2 episode at both sites and in ATMo-Lab with cold
temperatures (Table 1 and Table S4, Fig. S3) which indicates that, in addition to long-range transported pollutants, local
traffic emissions were also trapped on the boundary and layer affected the air quality at the measurement site. Also, higher
concentrations of α-pinene with short atmospheric lifetime (~few hours, Hellén et al., 2012) indicated local influence. α-
pinene is not long-range transported, and sources are expected to be local/regional, α-pinene has also anthropogenic
sources related to human activity (e.g. cleaning and hygiene products). Episode E1 took place during the weekend so its
concentrations should be compared against the traffic weekend situation.

Concentrations of major gaseous and chemical components were measured also at the UB Supersite during the campaign
(Table S4). The concentrations of traffic related components PN, LDSA and $NO_2$ were on average 2–3 times higher at
the Traffic Supersite compared to the UB Supersite during non-episodic workdays which is expected to be due to the
much less influence of traffic at the UB Supersite. The same was noticed also for $PM_{2.5}$ concentrations between the two
sites indicating its local source at the Traffic Supersite. Concentrations of these compounds were similar or only slightly
lower during non-episodic weekends at the UB Supersite so it can be concluded that the measured concentrations at the
UB Supersite correspond to urban background concentrations in the Helsinki area.
Concentration of organics and secondary inorganics showed similar increase during the three episodes at the UB Supersite
and Traffic Supersite (Fig. S6). The increased $PM_{2.5}$ concentration during the episodes is connected to the formation of
secondary particulate matter together with the increased concentration of organics. During non-episodic periods the
concentration of organics was higher at the Traffic Supersite compared to the UB Supersite indicating that it had a local
source, most probably traffic, at the Traffic Supersite.
During all episodes, the concentration of $PM_{2.5}$ and LDSA (Fig. S2, Table 1 and S3), and BC (Traffic Supersite and UB
Supersite, Fig. 6) showed clear increase in their concentrations. Also biomass burning tracer levoglucosan and the sum
of PAHs (Traffic Supersite) increased during the episodes. It seems that the long-range or regional transported air masses
contained of particles originated form biomass burning and that the transported particles clearly increased the $PM_{2.5}$ and
BC concentrations and so degrading the air quality in Helsinki area. During cold season wood burning for heating
purposes especially in detached house areas in Helsinki takes place. However, the effect of local wood burning on
pollutant concentrations at the Traffic Supersite is minimal on annual level compared to the effect of traffic related
pollutants (Aurela et al., 2015; Helin et al., 2021; Kangas et al., 2024). This can be seen also from the relatively low
concentration of levoglucosan and sum of PAHs during non-episodic period. However, the concentration of levoglucosan
and PAHs increased during the episodes indicating that long-range or regionally transported biomass burning aerosol was
transported to the measurement site.

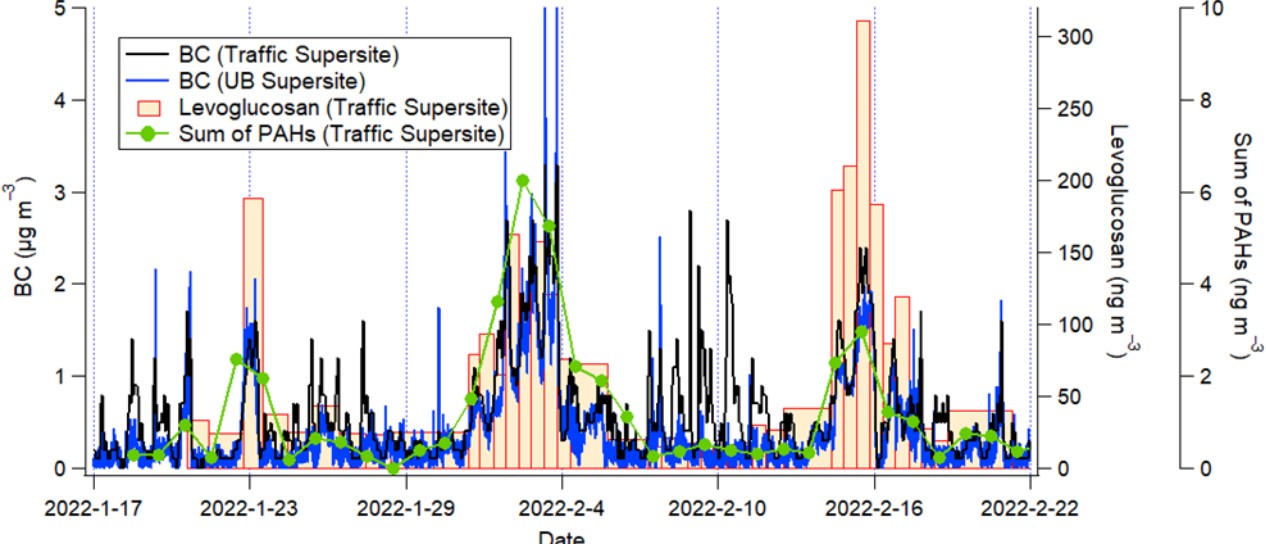


**Figure 6.** Concentrations of BC at the Traffic Supersite and at the UB Supersite as well as the concentrations of levoglucosan and the sum of measured PAH compounds at the Traffic Supersite, analysed from the filter samples. The filter sample times are the mean of sampling start and stop times. The PAH sampling time during 15 February consisted only 12 hours.

Fig. S7 shows the particle number size distributions at the Traffic Supersite and UB Supersite during the three episodes and during the traffic related (non-episodic) period. The mean particle number concentration did not show any marked increase during the episodes compared to the non-episodic situation at the Traffic Supersite (Table 1), but the mean LDSA concentrations measured with AQ Urban instrument were clearly higher. The increased LDSA concentration during the episodes is connected to higher concentration of larger particles at both sites (Fig. S7). The higher LDSA concentration at the Traffic Supersite compared to the UB Supersite during non-episodic situation is due to the higher concentration of traffic related ultrafine particles at the Traffic Supersite.

HYSPLIT back trajectories were calculated during these episodes on daily basis (3-hour resolution, 96-hour back trajectories, Fig. S8). The air masses during the short period E1 came from the Arctic areas. The air masses during the E2 period between 31 January and 3 February came from Eastern Europe (Moscow area, Fig. S8) straight to the measurement site and between 4 February and 5 February came across Baltia and Belarus. During the E3 period the airmasses circulated first over Central Europe (Poland and Baltia, 13–14 February) and then arrived from Southern Europe over Romania, Ukraine, Belarus, Poland and Baltia.

The coldest temperatures during the measurement campaign were measured during periods E1 and E2 (Fig. 2) and only during the period E3 the temperature was near 0 $°$C. It is possible that especially traffic related particulate components and trace gases accumulated in the boundary layer during the cold days and that the increased concentrations are due both this accumulation and long-range transported pollutants. The concentrations of traffic related gaseous pollutants CO, $CO_2$, NO, and $NO_2$ showed higher concentrations at the Traffic Supersite (Table 1) during episodes compared to non-episodic period, especially during the episode E2 when the wind speed was very low at the last days. For $CO_2$ this increase was minimal and for the PN concentration only slight difference between episodes and workdays without episodes could be seen. The difference of PN concentration between non episodic workdays and weekends was clearer (Table 1).

3.1.6      Source apportionment of organics in aerosol particles

A source apportionment of organic aerosols was conducted on the AMS data collected at the Traffic Supersite. PMF solution consisted of 6 factors: OA with a significant signal at m/z 60 ($C_2H_4O_2^+$) and 61 ($C_2H_5O_2^+$), Tr-OOA; low-volatility oxygenated OA (LV-OOA) with a large signal at m/z 44 ($CO_2^+$); hydrocarbon-like OA (HOA) mostly composed of $C_XH_Y^+$ fragments; biomass burning OA (BBOA) with characteristic m/z 60 ($C_2H_4O_2^+$) and m/z 73 ($C_3H_5O_2^+$) signal peaks; semi-volatile oxygenated OA (SV-OOA) with high signal at m/z 43 ($C_2H_3O^+$), and LV-OOA-BB that had also a high signal at m/z 44 but as well a significant signal at m/z 60 (Fig. S9 and S10). LV-OOA represents primarily regional or long-range transport emissions, while the pronounced m/z 60 signal in LV-OOA-BB strongly indicates its biomass burning origin. In contrast, the exact source of Tr-OOA remains uncertain. It is likely linked to vehicular emissions based on its mass spectra and a diurnal profile that closely aligns with HOA, the factor representing primary traffic-related OA. The PMF results have been shown earlier in Barreira et al. (2024) which studied the light absorption characteristics of organics.

The share of chemical composition of major measured compounds (BC, organics, sulphate, nitrate, ammonium, and
chloride) as well as the share of organic factors are shown in Fig. S11. Concerning primary emissions, HOA was the
dominant factor at the Traffic Supersite, with an average contribution of 12.5 % during the whole campaign (Fig. S11).
In fact, HOA had a moderate correlation with $NO_x$ ($R^2$ equal to 0.74) and its concentration was particularly high during
workdays and traffic rush hours (Table 1 and Fig. S11). On the other hand, the overall contribution from BBOA was small
(6.1%), even though the mean concentration of BBOA was like the one of HOA during weekends. This low BBOA
contribution reflects the fact that residences around the measurement site do not use biomass burning as a dominant
heating source (e.g. Kangas et al., 2024 and Barreira et al., 2021) The obtained results agree with previous observations
at the same measurement site concerning primary organic aerosol chemical characteristics during wintertime (Lepistö et
al., 2023b).
Contrary to primary emissions, SV-OOA, LV-OOA, and LV-OOA-BB, which mostly represent secondary organic
emissions, constituted on average 67.7 % of the total organics during the campaign period (Table 1 and Fig. S11). The
concentration of LV-OOA-BB factor was especially high during the E1, E2 and E3 periods reaching up to 52.8, 47.4, and
56.2% for the forementioned periods, respectively, indicating a strong influence of long-range transported aerosol during
these periods. These results reveal the high importance of long-range transport episodes to the atmospheric particulate
mass and composition in Helsinki. Furthermore, they suggest a heterogenicity of chemical and physical properties of
long-range transported particles because of distinct production and emission sources. Interestingly, the SV-OOA
contribution was relatively constant during all events (approximately 15%), except in E3 due to the increased contribution
from LV-OOA-BB (lower SV-OOA contribution). SV-OOA is expected to be low during winter due to decreased
atmospheric photochemistry comparatively to the warmest periods of the year (e.g. Praplan et al., 2017). During episodes
E2 and E3, all PMF factors exhibited heightened concentrations, including HOA known to mostly originating from local
traffic. This result suggests that, during these episodes, aerosol particles comprised a blend of both transported and locally
emitted pollutants, despite the dominance of LV-OOA aerosol.
3.1.7    Traffic related (non-episodic) period
Table 1 shows the mean traffic frequencies and concentrations of gaseous and particulate pollutants at the Traffic
Supersite during traffic related (non-episodic) period divided further as workdays and weekends and during the three
periods with high pollutant concentrations.
The traffic frequency during workdays was on average 1.4 times higher than during the weekends which is mostly due to
the lack of commuter traffic during weekends. Probably the number of trucks and other heavy-duty vehicles during the
weekends is also smaller. The mean concentrations of PN, NO, $NO_2$, BC, $PM_{2.5-10}$, and HOA were higher during the
workdays compared to weekends, which is expected since PN, NO, $NO_2$, BC, and HOA are emitted directly from the
motor engines. Local traffic also increases concentration of coarse particles, but their concentration also depends on
meteorology like rain, snow cover, or wind speed. Concentrations of most pollutants decreased with increasing wind
speed despite of the wind direction.
Mean LDSA concentration is connected to both PN concentration and particle size. The lower mean PN concentration
together with lower mean LDSA concentrations during non-episodic weekends indicates that LDSA concentration is
connected to local PN emissions during non-episodic period. The effect of long-range or regionally transported aerosol
to the LDSA concentration is clearly stronger than the effect of local traffic during the episodes. However, in general the
local traffic related PN emissions dominate the LDSA concentration at the Traffic Supersite which can be seen on the
higher LDSA concentration during the daytime at the Traffic Supersite compared to the UB Supersite (Fig. S12). The
slightly higher mean concentration of organics during the workdays is probably due to the higher HOA concentrations
during workdays.

The hourly variations of PN, BC, $NO_x$, $PM_{2.5}$, $PM_{2.5-10}$, and LDSA during non-episodic period are shown in Fig. S12
(workdays) and Fig. S13 (weekends) at the Traffic Supersite and UB Supersite station. During workdays the hourly
variations of PN, BC, $NO_x$ and LDSA are clearly connected to local traffic frequencies (Fig. 2 and Fig. S12) showing the
morning and late afternoon rush hours. The concentrations of $PM_{2.5}$ and $PM_{2.5-10}$ also increased during daytime but were
not as clearly correlated to morning and afternoon rush hours. Their concentrations started to rise during morning hours
and stayed high during the daytime. The hourly variations of these compounds were similar at the UB Supersite, but their
concentrations were much lower.

The hourly variations of total organics and the calculated factors HOA, BBOA, SV-OOA, LV-OOA, LV-OOA-BB, and
Tr-OOA during non-episodic period are shown in Fig. S14 (workdays) and Fig. S15 (weekends). The hourly variation of
HOA factor is clearly connected to traffic frequencies. In fact, its diurnal variation is similar to that of PN, $NO_x$ and BC
which are primarily from the engine emissions. During the weekends the diurnal pattern of HOA together with PN, $NO_x$
and BC show higher concentrations during afternoon and late evening following the diurnal variation of traffic
frequencies. The correlation coefficients ($R^2$) between HOA and NO and $NO_2$ were 0.67 and 0.74 respectively. The factor
connected to biomass burning (BBOA) shows two peaks; one in midday and another in evening. The evening peak is
probably connected to wood burning in Helsinki area. Wood burning takes places in detached houses in Helsinki in sauna
stoves and housewarming purposes especially during cold months. The evening peak of BBOA was clearly seen also
during weekends and it started already after late afternoon which is due to more active use of sauna stoves and fireplaces
during weekends. The diurnal cycle of Tr-OOA during workdays (Fig S14) indicates that it was connected to local traffic
related emissions. However, compared to HOA, its concentration did not clearly increase during afternoon and late
evening on weekends. The concentrations of oxidised organic factors SV-OOA and LV-OOA were quite similar during
the whole day, both on workdays and weekends. This indicates that their source was mostly of long-range or reginal
origin. Concentration of LV-OOA-BB factor was also very stable during the workdays but showed increased
concentrations during the evening on weekends. It is possible that the local or regional wood burning is shown in this
factor during the weekends.

3.1.8    Local pollution level comparison – CPCs and PDA
CPCs number concentration time series were used to compare local pollution level differences at the Traffic Supersite
and the UB Supersite stations. It can be seen from the time series (Fig. S16) that significantly higher particle concentration
levels are frequently observed at the Traffic Supersite during the campaign. To look at the differences between the sites
in more detail, the pollution detection algorithm was employed (PDA, Beck et al., 2022). The PDA identifies and flags
polluted periods in five steps, most importantly by the first filter step: the time derivative (gradient) of a concentration
over time.

PDA is primarily designed to identify and flag periods of polluted data in remote atmospheric composition time series,
but it might also be applied to locations where local contamination interference is so frequent that most data points exceed
the contribution from the underlying background in the period of interest, like in urban areas (Beck et al., 2022). PDA
only relies on the concentration time series datasets and is independent of ancillary datasets, such as meteorological
variables or black carbon data. Consequently, the PDA can provide valuable additional information on the pollution levels
and characteristics of urban conditions.

Fig. 7 and Fig. S17 show PDA filter results for the Traffic Supersite and the UB Supersite for the campaign period by
using typical PDA parameters for the 1 min time resolution data (Beck et al., 2022). CPC data gaps (Fig S17), 1.3 % and
6.6 % of the Traffic Supersite and UB Supersite, respectively, were assigned to one and removed for CPC total counts
(Table 2). It should be noted that although both CPCs were operational simultaneously most of the time during the
campaign, these data gaps cause some uncertainty in the analysis. Identical PDA settings were used for both stations.
Interquartile range (IQR) filter, instead of power law derivative filter was used as a first derivative filter as it is better
suited for the current (relatively polluted) urban data with no clear separation of data to polluted/unpolluted branches.
Two cases were compared, one with threshold filter for typical polluted concentration levels $> 10^4$ p cm$^{-3}$, (Fig. S17) and
with the upper threshold set to $3 \times 10^5$ p cm$^{-3}$ (upper concentration range of the measurements) to observe differences in
the derivative filtering without threshold filter that otherwise dominates in the urban environment (Fig 7). It can be seen
that the flagged (red) data fraction is more prominent in the Traffic Supersite data. All employed filter steps and PDA
parameters are shown in Tables S4 and S5.

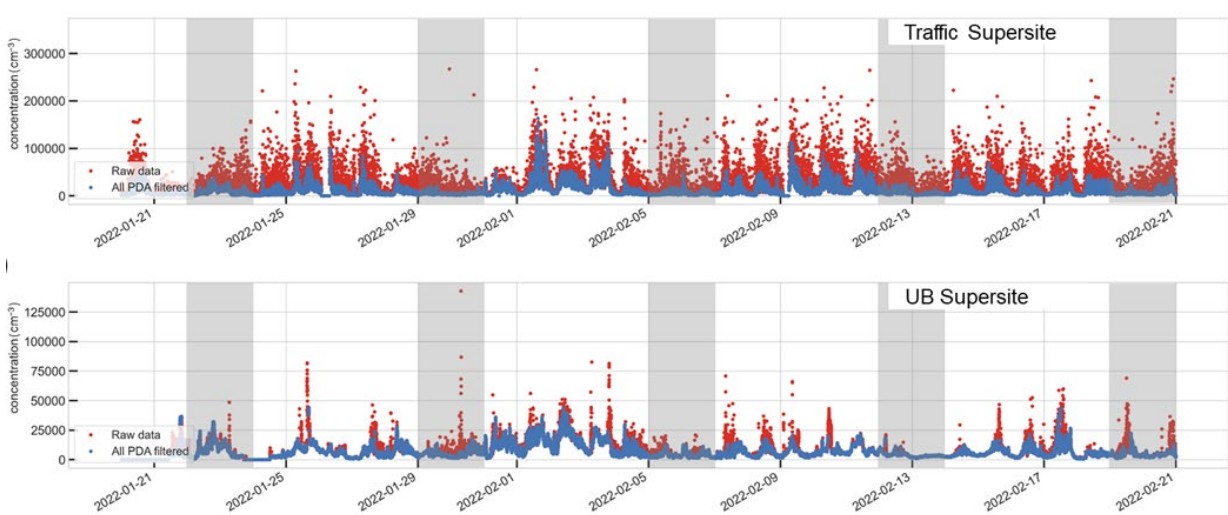


**Figure. 7.** PDA filter results for the Traffic Supersite (top) and the UB Supersite (bottom) after identical filtering steps, showing the
flagged data (red) with the threshold set to upper concentration range of the measurements. Shadowed periods represent weekends. See
Table S5 for the applied filter steps and parameters.

The PDA results are summarized in Table 2. In both cases, the ratio of the PDA filtered data for Traffic Supersite/UB
Supersite was similar (1.9 and 2.0). The higher pollution ratio for the case without the upper threshold filter (gradient
filter dominates), seems to reflect more polluted conditions due to local traffic and near roadside conditions at the Traffic
Supersite as the higher derivatives are representing periods of high concentration variability, i.e., due to local source
(Beck et al., 2022).

**Table 2.** An overview of the PDA results and percentage of data declared as polluted with the applied filtering steps.

| | Traffic Supersite (number of data points) | UB Supersite (number of data points) | Traffic Supersite (%) | UB Supersite (%) | Ratio Traffic Supersite/UB Supersite |
|---|---|---|---|---|---|
| Total counts (number of data points). | 46915 | 44383 | | | |
| PDA polluted (10 000 p cm$^{-3}$ threshold), Figure S17. | 33023 | 16463 | **70.4** | **37.1** | **1.9** |
| PDA polluted (no threshold, >300 000 p cm$^{-3}$), Figure 7. | 16638 | 7985 | **35.5** | **18.0** | **2.0** |

### 3.1.9 NAIS particle size distribution comparison
Fig. 8 presents hourly median particle size distributions obtained during the workdays (Monday–Friday) with the NAIS
particle mode, negative polarity at the Traffic Supersite (Fig. 8, top) and at the UB Supersite (Fig. 8, bottom). It is clear
also from the NAIS measurements that the Traffic Supersite location has significantly higher median particle
concentrations throughout the day, starting from the early working hours (~ 6:00) and continuing to late evening (22:00).
Furthermore, it appears that during office hours the relatively high particle concentrations (~ $10^4$ p cm$^{-3}$) are observed
also at the sub-3 nm size range and lower size range of nucleation mode (3–5 nm), whereas at the UB Supersite similar
concentrations are observed only for particles > 5 nm. At the UB Supersite the sub-5 nm particle concentrations are also
rapidly decreasing after ~ 17:00, while at the Traffic Supersite the concentrations remain relatively high until ~ 22:00.

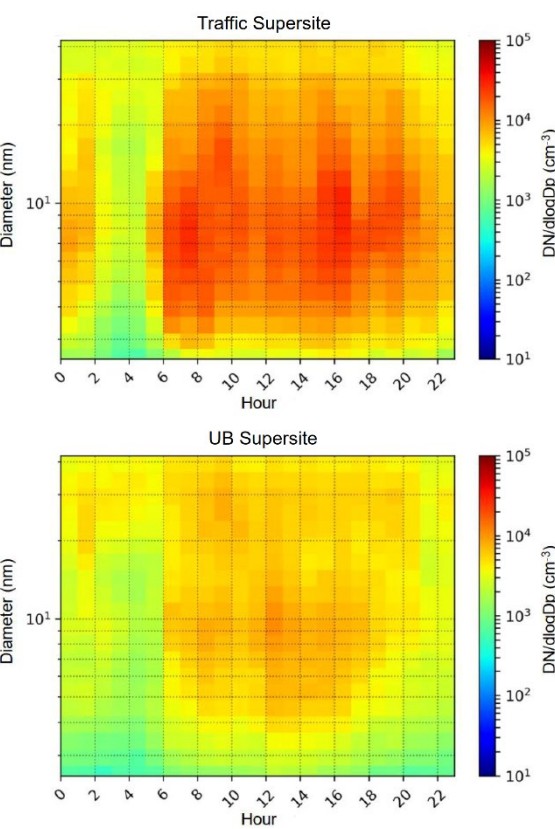


**Figure 8.** Hourly median particle size distributions obtained from NAIS negative polarity during the campaign workdays (Monday–Friday) at the Traffic Supersite (top) and the corresponding particle size distributions at the UB Supersite (bottom).

Data availability

Data described in this manuscript can be accessed on Zenodo repository under DOI 10.5281/zenodo.13254916 (Teinilä, 2024).

Code availability

The pollution detection algorithm described in Beck et al., 2022 is available on Zenodo at https://doi.org/10.5281/zenodo.5761101.

## 4    Conclusions

In this study physical and chemical properties of particulate matter and concentrations of trace gases were measured at an urban traffic site in Helsinki, Finland. The five-week intensive campaign took place at the Traffic Supersite in Helsinki in January–February 2022. The goal of the study was to characterise wintertime aerosol and obtain information on factors affecting air quality at urban traffic site in wintertime. To estimate the importance of local traffic and long-range transported pollutants on the air quality, measurements were made same time also at an urban background Supersite. A source apportionment of organics was performed for the SP-AMS measurements at the Traffic Supersite. The solution consisted of six factors of which three were connected to primary emissions (HOA, BBOA, and Tr-OOA) and three to aged aerosol (SV-OOA, LV-OOA and LV-OOA-BB).

During the intensive campaign the meteorological conditions such as temperature, snow cover, and rain varied largely. Three clear pollution episodes with elevated concentrations of particulate matter and trace gases took place during the campaign. During these episodes the increased pollutant concentrations were connected to trapping of local pollutants on the boundary layer, and long- and regional range transport of pollutants to the site. The concentrations of traffic related pollutants PN, NO, $NO_2$, BC, and HOA followed the traffic frequencies on an hourly basis and having also lower concentrations during weekends when traffic frequencies were lower. The source apportionment showed that, in addition to traffic related primary HOA, particulate matter consisted also biomass burning related aerosol (BBOA factor). During the pollution episodes high concentrations of secondary inorganics (sulphate, nitrate, and ammonium) and secondary organics (SV-OOA, LV-OOA, and LV-OOA-BB) were observed. Especially the concentration of secondary organics containing biomass burning material, LV-OOA-BB was very high, showing concentrations as high as 6 $\mu$g m$^{-3}$ and it was the dominant factor during episodes E1, E2 and E3. This together with the increased concentration of levoglucosan and BC indicate that long-range or regionally transported aerosol contained biomass burning originated particles.

The pollutant concentrations were affected also by meteorology like wind speed and temperature. During cold periods especially pollutants from local traffic trapped on the boundary layer increasing their concentrations. Stagnant conditions with low wind speed during coldest days inhibit the ventilation and removal of local pollutants effectively. Concentrations of most pollutants decreased with increasing wind speed.

It can be concluded that the air quality at the Traffic Supersite was affected by both changes in pollution sources and the removal of pollutants. The two most important pollution sources at the site were local traffic and long-range or regional transportation. Long-range or regional transported aerosols are present constantly in the Helsinki area, but we also observed episodes with markedly increased pollutant concentrations and increased $PM_{2.5}$ concentrations.  During these

episodes $PM_{2.5}$ mainly consisted of secondary inorganic and organic aerosol and black carbon. Trapping of pollutants with stagnant conditions during coldest days also increased pollutant concentrations originated from local traffic exhaust. As long-range transported pollutant episodes increased $PM_{2.5}$ mass, the pollutants from local traffic increased particle number concentration. The effect of local traffic on particle number concentration was most clearly seen in diurnal variation of PN with morning and afternoon rush hours and lowered PN concentration during weekends. A strong effect of traffic was seen also with the concentrations of the smallest nanoparticles (both < 5 nm and < 10 nm) which agrees with existing literature. As expected, due to traffic as a major local source, aromatic hydrocarbons made the highest contribution to the total measured concentration of SOA precursor VOCs (> C5).

The fact that we observed such a high contribution of long-range and regionally transported pollutants to PM mass, and the concentration of secondary inorganic and organic constituents show the need to tackle atmospheric pollutants not only at local level but in concerted actions involving regional and international regulative entities.

Competing interests

At least one of the (co-)authors is a member of the editorial board of Atmospheric Chemistry and Physics.

Acknowledgments

Long-term research co-operation and support from HSY to this project is gratefully acknowledged. Katja Moilanen from the City of Helsinki is acknowledged for the traffic count data. Financial support from Black Carbon Footprint project funded by Business Finland and participating companies (Grant 528/31/2019), from Technology Industries of Finland Centennial Foundation to Urban Air Quality 2.0 project, EU Horizon 2020 Framework Programme via the Research Infrastructures Services Reinforcing Air Quality Monitoring Capacities in European Urban & Industrial AreaS (RI-URBANS) project (GA-101036245) and Academy of Finland project BBrCaC (grant no. 341271) as well as Flagship ACCC (grant no. 337552, 337551) are gratefully acknowledged. This work was supported by the Finnish Research Impact Foundation under grant 4708620.

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
