# Peer review of "MEASUREMENT REPORT: WINTERTIME AEROSOL"

_EGUsphere, 2024_

## Author Comment (AC1)

Referee comments

We thank the Referees for their valuable comments on our manuscript. We made a revision to the manuscript based on the comment of the Referees, and we think that it improved the quality of the manuscript significantly.

**Referee #1**

The manuscript reports a comprehensive study of urban air quality in Helsinki, Finland. The work was carried out at two supersites using a bunch of instrumentation: traffic pollutant dominated measurement site and urban background station less affected by traffic. Interestingly, the urban station is less that one hundred meters further from the major traffic roads than the traffic station, but the observations were quite different. I have a few minor comments before its final publication.

Throughout the manuscript, measurement uncertainties should be added.

**Answer:** Information about the measurement uncertainties has been added to the experimental section for the instruments whose data has been shown in this paper. The paragraphs below are added at the end of each instrument section.

AMS

The uncertainties of the AMS measurement arise from several factors. One source of uncertainty is the used effective nitrate response factor which is determined by the calibration of the AMS. Also, the use of default RIE for the calculation of total OA concentration is a source of uncertainty as a single RIE value for organics may not represent thousands of different organic compounds found in particles. The fact that the lower size range of the AMS is 50 nm has only a minor effect on the measured concentrations since the majority of PM1 mass is in particles above this size. The calculation of CE based on the chemical composition of measured aerosol is an additional source increasing the uncertainty as it uses nitrate, ammonium and sulphate concentrations in the calculation. The overall uncertainty of the AMS measurements can be estimated to be about 20–30 %.

ACSM

The uncertainties related to the ACSM measurements are like those described for the AMS above. However, the measured concentration of chloride and ammonia were very low during the campaign so especially for these two components the measurement uncertainty is clearly higher. The estimation for the uncertainties is 50 % for ammonia and >50% for chloride.

GC-MS/FID

The detection limits for the different VOCs varied between 0.2 and 16 ng m$^{-3}$. Average uncertainty was 2.8, 25 and 18 ng m$^{-3}$ for terpenoids, aromatic compounds and C6-C15 alkanes, respectively.

NAIS

The total particle concentrations measured by the NAISs have been observed within ±50 % of the reference CPC concentration at 4–40 nm sizes (Asmi et al., 2009).

CPCs

The uncertainties of the CPCs are typically within 10% concentrations of the ambient aerosol ranging from a few thousands up to 100 000 particles cm$^{-3}$ (Schmitt et al., 2020). In both CPC types, butanol (n-Butyl alcohol) was used as a working fluid and data was collected at 1 min time resolution.

MAAP

At traffic Supersite the measured BC concentration was most of the time above the detection limit of the MAAP so the measurement uncertainty is mostly due to the uncertainties in the sampling like particle losses in the sampling lines. The uncertainty of the MAAP results can be estimated be around 10–15 %. At UB Supersite the measured BC concentration was more frequently near or below the detection limit. This can cause larger uncertainties for the BC measurements at UB Supersite.

AQ Urban

The measured LDSA concentration was typically above the detection limit of the AQ Urban instrument (1 $\mu m^2$ $cm^{-3}$).

DMPS

The particle number size distributions from 20 to 200 nm determined by the mobility particle size spectrometers are typically within an uncertainty range of around ±10%, while below and above this size range the uncertainty increases. For particle sizes above 200 nm, 30% uncertainty has been reported (Wiedensohler et al., 2012).

Picarro

The measured $CO_2$ and $CH_4$ concentrations were above the detection limit of Picarro at both sites. The uncertainty of these two gases is low (10 %) but for the measured CO concentrations the uncertainty can be larger.

EC/OC analyses

The uncertainties of the EC/OC analysis are ~15 % (Cavalli et al., 2023).

Line24-27, long sentence, please reword it.

**Answer**: The sentence has been reworded.

L94-95, Please add references for the mentioned figures.

**Answer:** The references have been added for the mentioned figures:

**Figure 1.** Stationary measurement locations and the driving route of the Aerosol and Trace-gas mobile laboratory. The Traffic Supersite is showed in the map as an orange balloon with a figure right bottom. The side street is showed as a blue balloon and the figure is right middle. The UB Supersite is showed as a red balloon and with a figure right top.

Section 2.2.1 Please make it clear if the intracavity Nd-YAG laser was used in the AMS. If laser was used, the RIE for BC should be mentioned too. How well does AMS-derived BC compare to other BC results?

**Answer:** The intracavity Nd-YAG laser was used in the AMS and this detail has been added to text. However, refractory BC (rBC) measured by the SP-AMS was not presented in the paper. Sentence clarifying this has been added to the manuscript:

However, rBC concentrations are not shown in this paper since the BC size emitted by traffic is partially below the transmission efficiency of the SP-AMS.

Although the rBC results are not presented in the paper, SP-AMS was calibrated for rBC. A RIE value is 0.09 for rBC was obtained. Also, the correlation between BC measured with MAAP and rBC measured with AMS was good ($R^2$=0.83) at Traffic Supersite. The slope between rBC (AMS) and BC (MAAP) was 0.66 which means that AMS gave lower rBC values compared to BC values measured with MAAP. It seems that the

measured rBC values measured with AMS are closer to BC values measured with MAAP during lower PN concentrations. Probably part of the traffic related BC particles is too small to be measured with SP-AMS.

L141, are you saying the transmission efficiency for the particles in the size range of 76-650 nm is 50%? This is not true, as the transmission efficiency is nearly 100% in the size range of 60 – 400nm.

**Answer:** It was said in Ng et al. (2011) that in the ACSM the transmission efficiency for the particles in the size range of 76-650 nm is 50% (Liu et al., 2007), however, it was not specified what kind of diameter (vacuum/aerodynamic?) it was. For clarity, the sentence has been rewritten:

which exhibits nearly 100% transmission efficiency from approximately 70 to 500 nm (aerodynamic diameter, e.g. Canagaratna et al., 2007; Jayne et al., 2000).

Since AMS and ACSM use similar aerodynamic lenses, the text above was moved to the paragraph describing the SP-AMS operation and in the ACSM chapter it is said only:

that is similar to the aerodynamic lens used in the AMS.

Line 163, consist->consists

**Answer:** Word has been corrected as suggested by the referee.

Line195-204, are the data used anywhere in the manuscript?

**Answer:** CI-API-TOF-MS data is not shown in this paper, and this is now mentioned in the manuscript.

Section 2.2.1. The section does a good job describing the instrumentations. For a quick grab of the idea of this section, it could also be nice to tabulate all the instruments used at the traffic superstation. You can also add additional instruments not listed in Table S2 at UB station.

**Answer:** The instruments at the Traffic Supersite have also been tabulated and the table has been added to the supplement.

L376, Add a period at the end of bracket.

**Answer:** Period has been added as suggested by the referee.

L385, it is unclear how the traffic frequencies were counted.

**Answer:** More detailed description about traffic frequency counting has been added to the manuscript:

Traffic frequencies are continuously counted by the City of Helsinki. Inductive loop sensors are installed below the asphalt surface for each driving lane. As the magnetic field of a vehicle passes over the inductive loop, it generates signals that are then recorded. The traffic frequencies were measured about 500 m from the station, but after the measurement point traffic directed to the city centre is divided into two other main streets before the Traffic Supersite. The average number of vehicles passing the Traffic Supersite during workdays was 17 000 per day which is about 40 % less than at the point where traffic frequencies were measured.

L410, which instrument was PN measurement from?

**Answer:** It was added that the PN measurements were carried out with a CPC with a cut-off ($D_{p50}$) of 5.4 nm.

L422 Is SP-AMS derived BC concentration comparable to MAAP results.

**Answer:** The BC concentrations measured with AMS and MAAP correlate well. However, the measured rBC concentrations with AMS are lower compared to those measured with MAAP probably due to the small size of traffic related particles containing black carbon (see the answer for the comment above).

L459, Please add reference(s) for your argument.

**Answer:** References have been added:

Hellén, H., Praplan, A. P., Tykkä, T., Helin, A., Schallhart, S., Schiestl-Aalto, P. P., Bäck, J., and Hakola, H.: Sesquiterpenes and oxygenated sesquiterpenes dominate the VOC ($C_5$–$C_{20}$) emissions of downy birches, Atmos. Chem. Phys., 21, 8045–8066, https://doi.org/10.5194/acp-21-8045-2021, 2021.

Hakola H, Taipale D, Praplan A, Schallhart S, Thomas S, Tykkä T, Helin A, Bäck J and Hellén H (2023): Emissions of volatile organic compounds from Norway spruce and potential atmospheric impacts. Front. For. Glob. Change 6:1116414. doi: 10.3389/ffgc.2023.1116414.

L472, can you guess the sources of a-pinene? In addition to the background concentrations, is it from regional or long-range transport?

**Answer:** Information about the sources of a-pinene has been added to the manuscript:

Due to the short lifetime of $\alpha$-pinene it is not long-range transported, and sources are expected to be local/regional. $\alpha$-pinene has also anthropogenic sources related to human activity (e.g. cleaning and hygiene products).

L531, ...are due to..

**Answer**: We think that the Referee is referring here to the sentence: "The higher LSDA concentration... is due to the higher concentration" so we think that the sentence is correct, and no changes were made.

Section 3.1.6 PMF did a nice job in distinguishing six organic factors. A few more words describing the Tr-LVOOA, LVOOA and LVOOA-BB would also help to understand these three factors and their sources.

**Answer:** More description of the PMF factors has been added:

A source apportionment of organic aerosols was conducted on the AMS data collected at the Traffic Supersite. PMF solution consisted of 6 factors: OA with a significant signal at m/z 60 ($C_2H_4O_2^+$) and 61 ($C_2H_5O_2^+$), Tr-OOA; low-volatility oxygenated OA (LV-OOA) with a large signal at m/z 44 ($CO_2^+$); hydrocarbon-like OA (HOA) mostly composed of $C_xH_y^+$ fragments; biomass burning OA (BBOA) with characteristic m/z 60 ($C_2H_4O_2^+$) and m/z 73 ($C_3H_5O_2^+$) signal peaks; semi-volatile oxygenated OA (SV-OOA) with high signal at m/z 43 ($C_2H_3O^+$), and LV-OOA-BB that had also a high signal at m/z 44 but as well a significant signal at m/z 60 (Fig. S9 and S10). LV-OOA represents primarily regional or long-range transport emissions, while the pronounced m/z 60 signal in LV-OOA-BB strongly indicates its biomass burning origin. In contrast, the exact source of Tr-OOA remains uncertain. It is likely linked to vehicular emissions based on its mass spectra and a diurnal profile that closely aligns with HOA, the factor representing primary traffic-related OA. The PMF results have been shown earlier in Barreira et al. (2024) which studied the light absorption characteristics of organics.

**Additional corrections**

1. The one-hour averages have been recalculated due to some mistakes in the start and end times of the measured components. The error in one-hour average data did not cause any meaningful changes to the average values shown in Table 1 and Table S3 or the correlation coefficients mentioned in text. However, these small changes have been corrected. The corrected data has been uploaded to Zenodo.

2. The text has been read throughout, and several long sentences have been shortened.

3.   The Abstract has been changed so that no abbreviations are used.

---

## Author Comment (AC2)

**Referee #2**

**General Comments**

The study's goal of examining how cold conditions affect pollutants in Helsinki, Finland, using a variety of monitoring techniques, was evidently investigated. A comprehensive and well-rounded research plan is demonstrated by the use of two measurement sites and the integration of diverse methodologies such as source apportionment and particular analysis algorithms. Even though it describes the pollution events that were observed and their possible causes, it could do a better job of highlighting the unique viewpoints and contributions that this specific study adds to the body of knowledge already available on urban air pollution in the winter.

**Specific Comments**

1. It would be beneficial to mention if there was any specific reason for choosing that particular "five-weeks" in January–February 2022 or if it was a random selection. Also, details about how representative that period is for wintertime conditions in Helsinki could strengthen the study's context.

   **Answer:** There was a reason to select those weeks for the campaign as that period typically represent Finnish winter conditions with low temperature and minimum sunlight. It was already mentioned in the article that "The aim of the study was to investigate the role of wintertime conditions in aerosol formation and precursor gases, black carbon emissions, emission sources, and their influence on particles' physical and chemical properties."

   We added to manuscript (Introduction) also:

   During wintertime, temperature inversion episodes cause traffic related pollutants to be trapped on the boundary layer hindering the mixing and dilution of pollutants. Also, photochemical reactions are minimal during wintertime and the contribution of biogenic emissions is limited.

   And to section 3.1.1 (Meteorology):

   The conditions during the winter campaign (temperature, inversion episodes and variable snow cover) represented typical winter conditions in Helsinki.

2. The authors need to clearly explain the reason for restricting the mobile measurements (ATMo - Lab) to the daytime between 6:30 and 19:30. Since air pollution patterns can vary significantly between daytime and nighttime, and different sources might have different activity levels during these periods, it is essential to know how the exclusion of nighttime measurements might affect the source apportionment and understanding of overall pollution dynamics.

   **Answer:**

   The focus of ATMo-Lab measurements was to understand how the effects of road traffic vary at the studied street canyon compared to more-open-sections of the same road. Also, the aim was to study the dispersion of road traffic emissions to the adjacent streets in a built environment, which is very relevant for air quality in urban environments in general. Therefore, the measurements were conducted only between 6:30 and 19:30, because the effects of traffic were the clearest during that time. It is true that nighttime activity and conditions differ from daytime, and, thus, the ATMo-Lab measurement cannot be utilized in a comprehensive source

apportionment analysis like the stationary measurement data. Also, for a more practical reason, nighttime measurements with the ATMo-Lab at the studied location were not possible due to available resources.

We added to the manuscript:

The focus of the ATMo-Lab measurements was to understand how the effects of road traffic vary in the studied street canyon compared to more open sections of the same road. Also, the aim was to study the dispersion of road traffic emissions to the adjacent streets in a built environment.

and

... because the effects of traffic were clearest during that time. Measurement setup inside the Aerosol and Trace-gas mobile laboratory ...

3. The authors must clearly describe the specific calculation method employed for determining the mixing height. Whether it's based on a particular theoretical model (e.g., a thermodynamic model, a boundary layer parameterization method, etc.), an empirical formula, or a combination of both, full relevant data and details should be provided.

**Answer:** More detailed description of the mixing height calculation has been added to the supplement.

4. The suggestion to incorporate a wind rose diagram at Fig. 2 has the potential to enhance the comprehensiveness and interpretability of the presented data.

**Answer:** The wind rose diagram has been added to Fig. 2 with the mention that the wind speed and wind direction were measured in Pasila weather station about 1 km distance from the Traffic Supersite station. The wind rose diagram gives an overall view of the prevailing wind direction. However, since the Traffic Supersite station is at a street canyon, the nearby tall buildings may alter the wind direction along the street lanes. Additionally, the wind rose plotted from the Kumpula weather station data is added. The Kumpula weather station stands next to the UB Supersite station.

[Figure]

5. The omission of standard values for the pollutant concentrations presented in lines 400 - 407. Without these reference values, it becomes extremely difficult for readers to assess the severity and significance of the measured pollutant levels. In addition to local standards, providing world

or international standards (such as those set by organizations like the World Health Organization for certain key pollutants) would offer a broader perspective.

**Answer:** The WHO reference values of $PM_{2.5}$, $PM_{10}$, $NO_2$ and $O_3$ have been added as suggested by the Referee.

6. Apart from summarizing the primary elements derived from the source analysis, it is advised to conduct additional analysis on the variations in the contributions of various sources in various pollution incidents and time periods (e.g., weekdays versus weekends, and daytime versus nighttime), as well as to thoroughly investigate the dynamics of the sources and their connections with weather and traffic patterns.

**Answer:** More discussion about the variations of PMF factors are added to the text:

The hourly variations of total organics and the calculated factors HOA, BBOA, SV-OOA, LV-OOA, LV-OOA-BB, and Tr-OOA during non-episodic period are shown in Fig. S14 (workdays) and Fig. S15 (weekends). The hourly variation of HOA factor is clearly connected to traffic frequencies. In fact, its diurnal variation is similar to that of PN, $NO_x$ and BC which are primarily from the engine emissions. During the weekends the diurnal pattern of HOA together with PN, $NO_x$ and BC show higher concentrations during afternoon and late evening following the diurnal variation of traffic frequencies. The correlation coefficients ($R^2$) between HOA and NO and $NO_2$ were 0.67 and 0.74 respectively. The factor connected to biomass burning (BBOA) shows two peaks; one in midday and another in evening. The evening peak is probably connected to wood burning in Helsinki area. Wood burning takes places in detached houses in Helsinki in sauna stoves and housewarming purposes especially during cold months. The evening peak of BBOA was clearly seen also during weekends and it started already after late afternoon which is due to more active use of sauna stoves and fireplaces during weekends. The diurnal cycle of Tr-OOA during workdays (Fig S14) indicates that it was connected to local traffic related emissions. However, compared to HOA, its concentration did not clearly increase during afternoon and late evening on weekends. The concentrations of oxidised organic factors SV-OOA and LV-OOA were quite similar during the whole day, both on workdays and weekends. This indicates that their source was mostly of long-range or reginal origin. Concentration of LV-OOA-BB factor was also very stable during the workdays but showed increased concentrations during the evening on weekends. It is possible that the local or regional wood burning is shown in this factor during the weekends.

7. Validate and supplement the source analysis results with other research methods (e.g., emission inventory data, PCA, etc.) to enhance the credibility and persuasiveness of the results.

**Answer:** We acknowledge this comment from the Referee, but we think that this topic is not in the scope of this article. The validation of source analysis results with other research methods is important and therefore requires a separate article to be done thoroughly.

8. The article has relatively little coverage on the chemical transformation mechanisms of pollutants during long-range transport. Research in this aspect should be strengthened. For example, by measuring the concentration changes of relevant precursors and products during the transport process, combined with the simulation of atmospheric chemistry models, analyze the generation processes and rates of secondary pollutants (such as sulfates, nitrates, secondary organic aerosols, etc.), and clarify the main chemical transformation reactions at different transport stages, so as to evaluate the impact of transport on the chemical composition and properties of pollutants more comprehensively.

**Answer:** Regarding the chemical transformation mechanisms of pollutants during long-range transport, there are huge variations in the lifetimes of the studied VOCs. While the lifetime of benzene is several months, some of the terpenoids (e.g. limonene) are oxidized within a few hours. Therefore, the longer living VOCs are expected to accumulate in the long-range transported air

masses. Also, shorter living α-pinene was higher than the average during the episodes indicating impact of also more local/regional sources. Chemical transformation will be studied in follow-up papers.

In terms of secondary aerosol components sulfate, nitrate, ammonium and SOA, are mostly formed from gaseous $SO_2$, $NO_X$, and VOCs in the atmosphere. During the episodes of long-range transport emissions (E1-E3), their concentrations were clearly larger than during the traffic periods (Table 1). For the oxygenated OOA components, SV-OOA and LV-OOA, the increase during the episodes was not so clear, however, the concentration of LV-OOA with BBOA was much larger during the episodes than during traffic periods suggesting its being log-range transported. The analysis of the generation processes and rates of secondary pollutants, and the main chemical transformation reactions at different transport stages have not been studied in detail in this paper as its out of the focus of article. However, as it is an important aspect, it could be studied in later papers.

9.  For regional transport, insufficient attention has been paid to the impact exerted by the terrain and the characteristics of the underlying surface in the vicinity of the measurement sites on the transport of pollutants.

    **Answer:** There was already some information about the effect of terrain in the text:

The $PM_{2.5–10}$ concentration was relatively low during the campaign. This is due to rainfall, snowfall, and snow covering the streets during the campaign which inhibited the formation and re-suspension of street dust. Most of the street dust is in coarse particle size, but it is in some degree also in fine particle size range. The lack of street dust episodes in winter explains, at least partly, why the mean $PM_{2.5}$ is also lower than that measured at the Traffic Supersite throughout in years 2015–2019 (Rönkkö et al., 2023b).

**Technical Corrections**

1.  The lack of uniformity in the font used for pictures throughout the paper detracts from the overall professionalism and polish of the work.

    **Answer:** The figures have been changed so that similar fonts are now used.

2.  Some of the sentences are longer and more complex in structure, which may cause some difficulty for readers to understand. For example: line 133–135.

    **Answer:** The sentence mentioned by the referee has been rewritten and the paper has been checked and rewritten to avoid long sentences.

**Additional corrections**

1.  The one-hour averages have been recalculated due to some mistakes in the start and end times of the measured components. The error in one-hour average data did not cause any meaningful changes to the average values shown in Table 1 and Table S3 or the correlation coefficients

mentioned in text. However, these small changes have been corrected. The corrected data has been uploaded to Zenodo.

2. The text has been read throughout, and several long sentences have been shortened.

3. The Abstract has been changed so that no abbreviations are used.